



# Enhancing SO₃ Hydrolysis and Nucleation: The Role of
# Formic Sulfuric Anhydride
**Rui Wang [a], Rongrong Li [a], Shasha Chen [a], Ruxue Mu [a], Changming Zhang [b],**
**Xiaohui Ma [c, *], Majid Khan [d], Tianlei Zhang [a, *]**
[a] *Shaanxi Key Laboratory of Catalysis, School of Chemical & Environment Science, Shaanxi*
*University of Technology, Hanzhong, Shaanxi 723001, P. R. China*
[b] *Shaanxi Key Laboratory of Catalysis, School of Mechanical Engineering, Shaanxi University of*
*Technology, Hanzhong, Shaanxi 723001, P. R. China*
[c] *School of Environmental Engineering, Henan University of Technology, Zhengzhou, Henan*
*450001, China*
[d] *College of Chemistry, Fuzhou University, 350116, Fuzhou, China*
## Abstract
Although the nucleation route driven by sulfuric acid ($H_2SO_4$) and ammonia ($NH_3$) primarily
dominates new particle formation (NPF) in the atmosphere, exploring the role of other trace species
on $H_2SO_4$-$NH_3$ system is crucial for a more comprehensive insight into NPF processes. Formic
sulfuric anhydride (FSA) has been observed in atmospheric environment and is found in abundance
in atmospheric fine particles. Nevertheless, its effect on $SO_3$ hydrolysis and NPF remain poorly
understood. Here, we studied the enhancing effect of FSA on gaseous and interfacial $SO_3$ hydrolysis
as well as its impact on $H_2SO_4$-$NH_3$-driven NPF occurring through quantum chemical calculations,
atmospheric clusters dynamics code (ACDC) kinetics combined with Born-Oppenheimer molecular
dynamics (BOMD). Gaseous-phase findings indicate that FSA-catalyzed $SO_3$ hydrolysis is nearly
barrierless. At an [FSA] = $10^7$ molecules·cm⁻³, this reaction competes effectively with $SO_3$
hydrolysis in the presence of $HNO_3$ ($10^9$ molecules·cm⁻³), HCOOH ($10^8$ molecules·cm⁻³) and $H_2SO_4$
($10^6$ molecules·cm⁻³) in the range of 280.0-320.0 K. At the gas-liquid nanodroplet interface, BOMD
simulations reveal that FSA-mediated $SO_3$ hydrolysis follows a stepwise mechanism, completing
within a few picoseconds. Notably, FSA enhances the formation rate of $H_2SO_4$-$NH_3$ clusters by over
$10^7$ times in regions with relatively high [FSA] at elevated temperatures. Additionally, interfacial
FSA⁻ ion has the ability to appeal precursor species for particle formation from the gaseous phase
to the water nanodroplet interface, thereby facilitating particle growth. These results present new
comprehensions into both the pathways of $H_2SO_4$ formation and aerosol particle growth in polluted
boundary layer.
**Keywords:** gas phase, atmospheric behavior, new particle formation, air pollution

---

* Corresponding authors. Tel: +86-0916-2641083, Fax: +86-0916-2641083.

e-mail: ztianlei88@l63.com (T. L Zhang) and mxhsdu@163.com (X. H Ma)



## 1. Introduction


Sulfuric acid (H$_2$SO$_4$) is an important atmospheric pollutant closely associated with new
particle formation (NPF) events and is recognized as a vital precursor in the process of converting
gases into particles. It facilitates the formation of sulfate aerosols and acid rain in diverse
environments, influencing cloud formation, precipitation and the Earth's radiation balance,
ultimately contributing to climate change (Yao et al., 2018; Venkataraman et al., 2001; Kumar et
al., 2024). Experimental (Couling et al., 2003; Reiner and Arnold, 1993; Bondybey and English,
1985) and theoretical studies (Feng and Wang, 2023; Kumar et al., 2024; Zhang et al., 2025) have
shown that atmospheric gaseous H$_2$SO$_4$ primarily forms via SO$_3$ hydrolysis (Sarkar et al., 2019; Tao
et al., 2018; Carmona-García et al., 2021). However, the likelihood of direct SO$_3$ hydrolysis in the
atmosphere is low due to the high activation energy associated with the process (Chen and Plummer,
1985). Introducing a second water molecule has been shown to significantly lower the activation
energy, making SO$_3$ hydrolysis more efficient (Morokuma and Muguruma, 1994). Further research
indicates that, besides water molecules, other species such as formic acid (Kangas et al., 2020),
oxalic acid (Yang et al., 2021), nitric acid (Long et al., 2022), H$_2$SO$_4$ (Wang et al., 2024) and
ammonia (Sarkar et al., 2019) exhibit even greater catalytic efficiency in promoting SO$_3$ hydrolysis
for H$_2$SO$_4$ formation. These findings provide valuable theoretical insights for understanding H$_2$SO$_4$
sources, particularly in regions where pollutant concentrations are notably elevated. Nevertheless,
further investigation is necessary to fully understand the SO$_3$ hydrolysis mechanism in areas with
high levels of specific pollutants, to better assess its behavior and effects under different atmospheric
conditions.
Carboxylic sulfuric anhydrides (CSAs) are a recently identified class of atmospheric
organosulfides, formed by the cycloaddition of SO$_3$ with organic carboxylic acids present (Fleig et
al., 2012). These CSAs exhibit strong acidity and can act as proton transfer bridges, potentially
influencing SO$_3$ hydrolysis and promoting the formation of H$_2$SO$_4$ in regions with high CSA
concentrations. Research indicates that the gaseous CSA concentration can reach 10$^7$ molecules·cm$^{-}$
$^3$ (Smith et al., 2020), creating conditions that may impact SO$_3$ hydrolysis. As the simplest CSA,
formic sulfuric anhydride (FSA) has been characterized using microwave spectroscopic (Mackenzie
et al., 2015). FSA is more acidic than formic acid and may facilitate proton transfer in the gaseous



hydrolysis of $SO_3$. However, its role in this process has not yet been explored. Besides, it has been
reported that the interfacial environment both initiates the organization and clustering of hydrophilic
groups and acts as an effective medium for various atmospheric reactions (Ma et al., 2020; Zhong
et al., 2019; Tan et al., 2022; Wan et al., 2023). Notably, proton transfer routes induced by interfacial
water molecules accelerate numerous atmospheric reactions taking place on aerosols and droplets
surfaces. These reactions typically proceed at accelerated rates and can differ from similar processes
in the gas phase or bulk water (Tang et al., 2024; Fang et al., 2024; Martins-Costa and Ruiz-López,
2024). Thus, it is essential to investigate whether FSA accelerates $SO_3$ hydrolysis at the gas-liquid
nanodroplet interface, as this could offer valuable insights into atmospheric chemistry and the
mechanisms driving particle formation

Additionally, new species generated from gas-phase reactions of $SO_3$ with trace substances (Li

et al., 2018; Liu et al., 2019) can also significantly influence the NPF process. For example, Li et al.
(Li et al., 2018) revealed that $NH_2SO_3H$, formed from the reaction of $SO_3$ with $NH_3$, not only
contributes directly to $H_2SO_4$-$(CH_3)_2NH$ cluster formation but also enhances the maximum rate of
NPF from $H_2SO_4$ and $(CH_3)_2NH$ by approximately twofold in heavily polluted areas with high
concentrations of basic substances. Similarly, Liu et al. (Liu et al., 2019) predicted that methyl
hydrogen sulfate (MHS), formed from the reaction of $SO_3$ with methanol, significantly impacts
$H_2SO_4$-$(CH_3)_2NH$ nucleation, particularly in dry regions with high alcohol concentrations. FSA,
produced from the reaction of $SO_3$ with HCOOH, contains the $-OSO_3H$ functional group and
exhibits a binding capability comparable to that of $H_2SO_4$ with nucleation precursors like $NH_3$. The
potential role of FSA in enhancing $H_2SO_4$-$NH_3$ nucleation in the atmosphere requires further
investigation to fully understand its contribution to NPF processes.

This work examined the catalytic effect of FSA on $SO_3$ hydrolysis and $H_2SO_4$-$NH_3$ nucleation

particle formation. Specifically, the catalytic effects of FSA on gaseous $SO_3$ hydrolysis were firstly
explored. Following this, the differences between the gaseous and interfacial reactions of FSA-
catalyzed $SO_3$ hydrolysis were evaluated using BOMD simulations. Subsequently, a qualitative
evaluation of FSA's nucleation capability was conducted through molecular dynamics (MD)
simulations. Finally, the atmospheric implications of FSA on particle formation were analyzed. This
study not only deepens our understanding of the impact of FSA on $SO_3$ hydrolysis but also provides
new molecular-level mechanisms for the contribution to $H_2SO_4$-$NH_3$ particle formation.



## 2. Computational Methods

**2.1 Quantum Chemical Details**. To investigate the impact of formic sulfuric anhydride (FSA) on gaseous $SO_3$ hydrolysis, the M06-2X/6-311++G(2*df*,2*pd*) computational method, as implemented in Gaussian 09 software (Frisch, 2009), was employed to analyze the geometric structures and vibrational frequencies of the relevant species. We also carried out the calculation of intrinsic reaction coordinate to conduct the connections between the transition states and their corresponding pre-reactive and post-reactive complexes. To enhance the reliability of the relative Gibbs free energies, single-point energies at the CCSD(T)-F12/cc-pVDZ-F12 level were calculated using the ORCA software (Neese, 2012).

The most stable structure of the $(FSA)_x(SA)_y(A)_z$ ($z \leq x + y \leq 3$) clusters were obtained by the following three steps. Initially, the ABCluster program (Zhang and Dolg, 2015) was utilized to randomly produce $n \times 1000$ initial isomers (where $n$ = 2 to 4) which were subsequently evaluated using the PM6 method via MOPAC 2016 (Partanen et al., 2016). Next, up to $n \times 100$ lowest-energy isomers were chosen and further refined using the method of M06-2X/6-31+G(*d,p*). Lastly, the top $n \times 10$ isomers were re-optimized at the M06-2X/6-311++G(2*df*,2*pd*) method level to ascertain their isomers with the lowest energy. The optimized structures and their Gibbs free energies are detailed in Fig. S10 and Table S6, respectively.

**2.2 Rate Coefficient Computations.** Rate coefficients for FSA-assisted $SO_3$ hydrolysis were calculated via two steps as follows. First, the VRC-VTST methodology (Zhang et al., 2023; Zhang et al., 2024) was applied using the Polyrate program (Meana-Pañeda et al., 2024) to calculate the rate coefficients under high-pressure conditions. Next, the Master Equation Solver for Multi-Energy Well Reactions (Glowacki et al., 2012) was engaged in computing the rate coefficients for FSA-assisted $SO_3$ hydrolysis across a temperature range of 280.0 to 320.0 K. To estimate the rate coefficients for the barrier less formation of pre-reactive complexes from the separated reactants, we applied the Inverse Laplace Transform (ILT) method (Kumar et al., 2021). In parallel, RRKM theory (Bao et al., 2016) was utilized to estimate the rate coefficients for the transition from the pre-reactive complex to the post-reactive complex through a transition stat. Details of the ILT methods and RRKM theory are provided in Part 1 and Part 2 in the Supplement, respectively.

**2.3 BOMD Simulations**. BOMD simulations were conducted with the CP2K program



(Hutter et al., 2014). The BLYP functional was applied to address exchange and correlation
interactions (Becke, 1988; Lee et al., 1988). Grimme's dispersion-corrected method (Grimme
et al., 2010) was employed to account for the dispersion interactions and effectively handle
weak   dispersion   effects.   The   Goedecker-Teter-Hutter   conservation   pseudopotentials
(Goedecker et al., 1996) were done by using Gaussian DZVP basis set (Phillips et al., 2005)
and an auxiliary plane wave basis, ensuring accurate treatment of both valence and core
electrons. The plane wave basis set was established with a 280 Ry energy cutoff, while the Gaussian
basis set cutoff was set at 40 Ry. A supercell side length of 15 Å was used in gas phase simulations
to eliminate periodic boundary conditions with step of 0.5 fs. For interfacial reactions, a water
droplet containing 191 water molecules was initially pre-optimized through BOMD simulation for
approximately 5.0 ps at 300 K. Subsequently, $SO_3$ and FSA were positioned at the gas-liquid
nanodroplet interface to perform the simulations over 10 ps. A supercell side length of 35 Å was
set for gas-liquid nanodroplet interface simulations to prevent periodic interactions between
neighbouring water droplets, using a step of 1.0 fs. In all simulations under the NVT ensemble, a
stable temperature of 300 K was maintained using the Nose-Hoover thermostat.
**2.4 Molecular Dynamics Simulation of Nucleation**. Complete nucleation pathway was
simulated using the GROMACS 2024.3 software (Abraham et al., 2024), employing the general
AMBER force field, a widely utilized approach for modelling molecular dynamics (Li et al.,
2024b; Cheng et al., 2025; Zhao et al., 2019). The electrostatic potential was computed at M06-
2X/6-311++G(2$df$,2$pd$) level and the restrained electrostatic potential charges were determined
using Multiwfn 3.8 (Lu and Chen, 2012). The AMBER parameter and coordinate files were
constructed using Sobtop (Lu, 2023) and Packmol (Martínez et al., 2009), respectively. The
simulation was performed within a cubic simulation box, each side measuring 200 Å in length.
Following energy minimization, the system was further simulated under the NVT and NPT
ensembles at 298 K for durations of 100 ps and 40 ns, respectively. The Berendsen pressure coupling
method (Berendsen et al., 1984) and the velocity rescaling thermostat (Bussi et al., 2007) were used
to regulate pressure and temperature, respectively. The system applied periodic boundary conditions
to mimic an infinite environment, with a 1 fs time step. The electrostatic and van der Waals
interactions were set with a 1.4 nm cutoff distance, and the Particle-Mesh Ewald method (York et
al., 1993) was implemented for long-range electrostatics. Bond lengths were restricted by the



LINCS algorithm (Hess et al., 1997) to preserve structural integrity during the simulation.
**2.5 Atmospheric Cluster Dynamics Code (ACDC) Model.** The ACDC (McGrath et al.,
2012) was employed to investigate cluster formation rates and growth mechanisms for
$(FSA)_x(SA)_y(A)_z$ clusters. The ACDC simulations were supplied with thermodynamic data,
which was derived from quantum chemical calculations performed by M06-2X/6-
311++G($2df,2pd$). Accounting for all potential collision and evaporation processes, the
following formulation represents the birth-death equations:
$$\frac{dc_i}{dt} = \frac{1}{2}\sum_{j<i}\beta_{j,(i-j)}C_jC_{(i-j)} + \sum_j\gamma_{(i+j)\to i}C_{i+j} - \sum_j\beta_{i,j}C_iC_j - \frac{1}{2}\sum_{j<i}\gamma_{i\to j}C_i + Q_i - S_i \qquad (1)$$

In the above equation, $c_i$ represents the concentration of $i$ cluster, while $\beta_{i,j}$ stands for the
collision rate between $i$ and $j$ clusters. The term $\gamma_{(i+j)\to i}\to i$ refers to the rate at which the larger $i+j$
cluster breaks down (or evaporates) into $i$ and $j$ clusters. Additionally, $Q_i$ accounts for any possible
external source of $i$ cluster. To consider the external losses of $i$ cluster, a coagulation sink coefficient
of $2 \times 10^{-2}$ s$^{-1}$ was used, aligning with values typically found in polluted environments (Liu et al.,
2021b). In ACDC, boundary clusters must be sufficiently stable, which allows them to continue
growing. Therefore, the clusters of $(FSA)_2\cdot(SA)_2\cdot(A)_3$, $(FSA)_1\cdot(SA)_3\cdot(A)_3$, $(SA)_4\cdot(A)_3$ and
$(SA)_4\cdot(A)_4$ were selected as the boundary clusters in the SA-A-FSA system.
**3. Results and discussion**
**3.1 The Hydrolysis of SO$_3$ Assisted by FSA**
The SO$_3$ hydrolysis with HCOOSO$_3$H (FSA) can initially occur via the interaction between
SO$_3$ (or FSA) and H$_2$O to form SO$_3\cdots$H$_2$O (or FSA$\cdots$H$_2$O) dimer. Subsequently, the SO$_3\cdots$H$_2$O
dimer collides with FSA, and the FSA$\cdots$H$_2$O dimer interacts with SO$_3$. The predicted relative Gibbs
free energies of SO$_3\cdots$H$_2$O was 0.8 kcal·mol$^{-1}$ at the CCSD(T)-F12/cc-pVDZ-F12//M06-2X/6-
311++G($2df,2pd$) level, which is nearly previously reported values (-0.2 to 1.0 kcal·mol$^{-1}$) (Long et
al., 2013; Long et al., 2012; Lv et al., 2019; Bandyopadhyay et al., 2017). As compared with
FSA$\cdots$H$_2$O, the binding free energy of SO$_3\cdots$H$_2$O is less stable by 2.6 kcal·mol$^{-1}$, which leads to
the equilibrium coefficient of FSA$\cdots$H$_2$O ($2.63 \times 10^{-18}$-$2.49 \times 10^{-19}$ molecules·cm$^{-3}$) (Table S2)
being at least 10 times larger than that of SO$_3\cdots$H$_2$O ($2.45 \times 10^{-20}$-$5.10 \times 10^{-21}$ molecules·cm$^{-3}$ within
280.0-320.0 K). Under the available concentrations ([FSA] = $1.0 \times 10^7$, [SO$_3$] = $1.0 \times 10^3$
molecules·cm$^{-3}$) (Liu et al., 2019), the concentration of FSA$\cdots$H$_2$O is $1.36 \times 10^6$-$6.80 \times 10^6$





molecules·cm$^{-3}$ within 280.0-320.0 K, which is $10^6$ times larger than that of SO$_3$···H$_2$O (Table S3).
Therefore, it is predicted that SO$_3$ hydrolysis with FSA predominantly take places via the collision
between FSA···H$_2$O and SO$_3$.
Starting from the FSA···H$_2$O + SO$_3$ reactants, an eight-membered ring pre-reactive complex
SO$_3$···H$_2$O···FSA (named as IM$_{SA\_FSA}$) was found and its Gibbs free energy relative to the isolated
SO$_3$, H$_2$O and FSA reactants was -2.0 kcal·mol$^{-1}$. In comparison to the previously reported neutral
(SO$_3$···2H$_2$O) and acidic complexes SO$_3$···H$_2$O···$X$ ($X$ = HNO$_3$, HCOOH, (COOH)$_2$ and H$_2$SO$_4$)
(Yang et al., 2021; Long et al., 2012; Torrent-Sucarrat et al., 2012; Long et al., 2013), the stability
of the SO$_3$···H$_2$O···FSA complex is notably enhanced by 0.2-2.7 kcal·mol$^{-1}$. This is because the
positive electrostatic potential (ESP) of the hydrogen atom in the FSA molecule (Fig. S4) is stronger
than those in H$_2$O and $X$ molecules, resulting in stronger intermolecular interactions of
SO$_3$···H$_2$O···FSA. Following the IM$_{SA\_FSA}$ complex, the reaction proceeds via TS$_{SA\_FSA}$, leading to
the H$_2$SO$_4$···FSA formation. For the FSA-catalyzed SO$_3$ hydrolysis, its Gibbs free energy barrier is
2.5 kcal·mol$^{-1}$, representing a reduction of 22.1 kcal·mol$^{-1}$ relative to the SO$_3$ hydrolysis without
FSA (Table S1). Moreover, it is also 1.0-4.0 kcal·mol$^{-1}$ lower in free energy barrier than those of
the SO$_3$ hydrolysis with H$_2$O, HNO$_3$ and H$_2$SO$_4$ (Table S1). Therefore, FSA is clearly more effective
than H$_2$O, HNO$_3$ and H$_2$SO$_4$ in decreasing the energy barrier for SO$_3$ hydrolysis. H$_2$SO$_4$···FSA is
an eight-membered ring complex, similar to H$_2$SO$_4$···$X$ complexes in the SO$_3$ hydrolysis with $X$.
The predicted free energy of H$_2$SO$_4$···FSA (-12.9 kcal·mol$^{-1}$) is lower by 10.9 kcal·mol$^{-1}$ compared
to that of the IM$_{SA\_FSA}$ complex. This indicates the thermodynamic favorability of FSA-assisted SO$_3$
hydrolysis.
The computed rate coefficients for the hydrolysis of SO$_3$ with and without FSA, H$_2$O and $X$
within 280.0-320.0 K are shown in Table 1. As observed at 298.0 K, the rate coefficient for the SO$_3$
hydrolysis with FSA ($k_{FSA}$) is $7.71 \times 10^{-11}$ cm$^3$·molecule$^{-1}$·s$^{-1}$, surpassing that of the uncatalyzed
SO$_3$ hydrolysis by a factor of $10^{12}$. Additionally, the value of $k_{FSA}$ at 298.0 K is larger by factors of
60.23 and 84.63 than those for the SO$_3$ hydrolysis with H$_2$O ($k_{WM}$) and HNO$_3$ ($k_{NA}$), respectively.
Similarly, within 280.0-320.0 K in Table 1, FSA can compete with HCOOH, (COOH)$_2$ and H$_2$SO$_4$
with the value of $k_{FSA}$ being larger by factors of 1.02-1.64 than those of $k_{FA}$, $k_{OA}$ and $k_{SA}$. These
findings indicate that the catalytic efficiency of FSA in SO$_3$ hydrolysis surpasses that of H$_2$O and
HNO$_3$, and is comparable to HCOOH, (COOH)$_2$ and H$_2$SO$_4$.





To consider a contribution of FSA on $SO_3$ hydrolysis, the rate ratios between FSA- and *X*-
catalyzed $SO_3$ hydrolysis reactions were calculated, as shown in Table S5. As observed, the $SO_3$
hydrolysis with $H_2O$ is more favorable than with FSA because the $[H_2O]$ ($10^{16}$-$10^{18}$ molecules·cm$^-$
$^3$) is significantly greater than [FSA] ($10^7$ molecules·cm$^{-3}$). When the acid catalysts $HNO_3$ ($10^9$
molecules·cm$^{-3}$), HCOOH ($10^8$ molecules·cm$^{-3}$) and SA ($10^6$ molecules·cm$^{-3}$) are considered, FSA
dominates over them within 280.0-320.0 K as the rate ratio $\nu_{WM}/\nu_X$ is greater than 1. This reveals
that the FSA-assisted reaction is indispensable in $SO_3$ hydrolysis within regions affected by FSA
pollution and can significantly promote the hydrolysis of $SO_3$ within 280.0-320.0 K.

### 219    3.2 FSA-Catalyzed $SO_3$ Hydrolysis at the Gas-liquid Nanodroplet Interface

Aqueous interfaces are widespread across Earth's atmosphere. (Li et al., 2024a; Zhong et al.;
2017; Sun et al., 2024; Gao et al., 2024; Dong et al., 2024). The gas-liquid nanodroplet interface
serves as a significant site for adsorption and reactions, potentially enhancing atmospheric reaction
rates and leading to the emergence of novel mechanisms. However, at the gas-liquid nanodroplet
interface, comprehensive understanding of the mechanism for FSA-assisted $SO_3$ hydrolysis was
lacking. Notably, during the 150 ns simulation, $SO_3$, the FSA molecule and the $SO_3$-FSA complex
were observed to reside at the interface for 35.8%, 46.3% and 40.5% (Fig. S5), respectively,
revealing that the presence of $SO_3$, FSA molecule and $SO_3$-FSA complex cannot be ignored at the
gas-liquid nanodroplet interface. To further investigate this prediction, we performed BOMD
simulations to assess the FSA-assisted hydrolysis of $SO_3$ at the gas-liquid nanodroplet interface.
Similar to the reactions of $SO_3$ with other acidic species at this interface, the interaction between
$SO_3$ and FSA at the aqueous interface might take place via three pathways: (*i*) direct interaction of
$SO_3$ with adsorbed FSA; (*ii*) interaction of adsorbed $SO_3$ with FSA; or (*iii*) reaction starting from
the $SO_3$-FSA complex. Given the high reactivity and the brief residency time of $SO_3$ and FSA at the
interface, as evidenced by their short lifetimes (Fig. S6) of only a few picoseconds and rapid
formation of SA$^-$ and FSA$^-$ ion, the simulations have primarily considered the model of (*iii*). This
focus enabled a deeper understanding of the interfacial dynamics and the mechanisms underpinning
these rapid transformations.
Unlike the gaseous hydrolysis mechanism of $SO_3$ with FSA, which occurs through the one-
step mechanism, interfacial $SO_3$ hydrolysis mediated by FSA occurs via a stepwise mechanism (Fig.
2, Fig. S7 and Movie S1), consisting of three steps: *i*) $SO_3$ hydrolysis along with proton transfer



outside the ring; *ii*) the deprotonation of FSA; and *iii*) the deprotonation of $H_2SO_4$. Specifically, at
0 ps, a loop-structure complex, $SO_3\cdots(H_2O)_2\cdots FSA$, was initially found with the formations of three
hydrogen bonds ($d_{(O6\cdots H4)}$ = 1.75; $d_{(O3\cdots H2)}$ = 1.92 and $d_{(O5\cdots H3)}$ = 2.39 Å) and a van der Waals
interaction ($d_{(O1\cdots S)}$ = 2.31 Å). Then, the loop structure mechanism proceeded along with the
simultaneous event of the proton transfer outside the ring. At 1.01 ps, an arrangement resembling a
transition state was found for the interfacial $SO_3$ hydrolysis, characterized by shortening of the S-
O1 and O2-H1 bonds and elongation of the O1-H1 bond. By 1.14 ps, the S-O1 and O2-H1 bond
lengths had reduced to 1.45 Å and 0.97 Å, respectively, while the O1-H1 bond had elongated to
1.42 Å, indicating the formation of $HSO_4^-$ and $H_3O^+$ ions. Due to the strong acidity of FSA, the H3
atom of FSA was moved to the O5 atom of the $HSO_4^-$ ion at 1.87 ps, leading to $H_2SO_4$ molecule and
$FSA^-$ ion. Finally, the deprotonation of $H_2SO_4$ was completed at 2.18 ps, with the H2 atom of $H_2SO_4$
moved to one interfacial water molecule inside the ring. In contrast to the $SO_3$ hydrolysis with FSA
in the gas phase, which does not proceed within 100 ps, the reaction at the gas-liquid nanodroplet
interface rapidly proceeds within just a few picoseconds. This indicates that interfacial water
molecules at the gas-liquid nanodroplet interface can accelerate the $SO_3$ hydrolysis.
Interestingly, the formation of $FSA^-$ and $HSO_4^-$ is highly stable, and their dissociation did not
occur within 10 ps. Species such as $H_2SO_4$ (SA), $NH_3$ (A), $HNO_3$, and $(COOH)_2$ are identified as
candidates for particle formation, with the SA-A cluster serving as a significant precursor to
atmospheric aerosols. Calculated binding free energies of the corresponding bimolecular clusters
were shown in Table 2 where the computed binding free energies agree well with previous values
(Zhong et al., 2019). As shown, the interactions of $FSA^-$-SA (-21.2 kcal·mol$^{-1}$) and $FSA^-$-$HNO_3$ (-
12.1 kcal·mol$^{-1}$) are stronger than that of SA-A (-8.9 kcal·mol$^{-1}$), illustrating that interfacial $FSA^-$
and $H_3O^+$ ions can attract precursor molecules from the gaseous phase to the aqueous nanodroplet
surface, and thus facilitating particle growth. Additionally, the enhancing potential of the $FSA^-$ ion
on the SA-A cluster was assessed by examining the binding free energies of the SA-A-$FSA^-$ and
SA-A-*Y* (*Y* = $HOOCCH_2COOH$, $HOCCOOSO_3H$, $CH_3OSO_3H$, $HOOCCH_2CH(NH_2)COOH$ and
$HOCH_2COOH$) clusters. The binding free energies of SA-A-$FSA^-$ and SA-A-*Y* clusters listed in
Table 2 were consistent with previously reported values (Rong et al., 2020; Zhang et al., 2018;
Zhang et al., 2017; Gao et al., 2023; Liu et al., 2021a). Notably, compared to SA-A-*Y*, the binding
free energy of SA-A-$FSA^-$ (-25.6 kcal·mol$^{-1}$) was larger than 5.2-12.8 kcal·mol$^{-1}$, indicating that the



FSA⁻ at the interface exhibits a greater nucleation capability than gaseous molecule *Y*. Consequently,
FSA⁻ is expected to demonstrate enhanced nucleation potential at the gas-liquid interface.

### 3.3 FSA's Role in Nucleation and Cluster Formation

Electrostatic potential (ESP) analysis was conducted to predict the potential hydrogen bond
binding sites among FSA, SA and A. The -OH moiety in the FSA molecule contains a highly
electrophilic hydrogen atom, making it a favorable donor site for hydrogen bonds (ESP value: +60.6
kcal·mol⁻¹) (Fig. 3). Meanwhile, the terminal oxygen atoms of the $-SO_3H$ and -COOH moieties in
FSA can act as an effective hydrogen bond receptor site due to their stronger electronegativity (ESP
values: -23.8, -22.4 and -13.0 kcal·mol⁻¹). Thus, FSA can form stable clusters by forming hydrogen
bonds with SA and A.
Using MD simulations, the aggregation behavior of FSA with SA and A molecules was
investigated at various atmospheric temperatures (Fig. 4 and Figs. S8-S9). In these simulation
systems, 5 FSA, 5 SA, 10 A, 20 $H_2O$, 41 $O_2$ and 154 $N_2$ molecules were included. Notably, the
complete stable $(FSA)_5 \cdot (SA)_5 \cdot (A)_{10}$ cluster was observed at all the three simulations temperatures.
With rising temperatures, the aggregation time for the formation of $(FSA)_5 \cdot (SA)_5 \cdot (A)_{10}$ cluster (Fig.
4(a)) increases. This observed phenomenon of aggregation implies that lower temperatures are more
conducive to form the $(FSA)_5 \cdot (SA)_5 \cdot (A)_{10}$ cluster. Fig. 4(b) displayed the snapshots of the nucleation
simulation at 258.15 K. The initial simulation at 0 ns shows that there is not effective nucleation, as
all molecules in the system are scattered (Fig. 4(b)). Subsequently, FSA can bind with SA and A to
form FSA·A, FSA·SA·A and $FSA·SA·(A)_3$ clusters at 1.5 ns, and then the FSA·SA·A,
$(FSA)_2 \cdot SA \cdot (A)_3$ and $(FSA)_2 \cdot (SA)_2 \cdot (A)_3$ clusters are formed at 3.0 ns. Next, with further aggregation
of FSA molecules, $(FSA)_2 \cdot SA \cdot (A)_4$ and $(FSA)_3 \cdot (SA)_3 \cdot (A)_4$ clusters are observed within 4.0 ns.
Finally, the FSA molecules fully aggregate to form $(FSA)_5 \cdot (SA)_5 \cdot (A)_{10}$ clusters at 7.5 ns, and this
complete cluster stays stable throughout the entire simulation period. It is noteworthy that the
numbers of FSA molecules can gradually interact with SA and A molecules to form relatively large
clusters, where hydrogen bonds among SA, A and FSA play a crucial role. Therefore, it is initially
predicted that FSA could act as a "participator" in NPF and could be directly involved in SA-A
nucleation. Further predictions regarding for the enhancement effect of FSA on SA-A molecular
clustering have been studied by considering the cluster stability, the formation rate and the growth
pathways.





### 3.4 The Impact of Atmospheric Conditions on the Thermodynamic Clusters Stability


The Gibbs free energies of formation ($\Delta G$, kcal·mol$^{-1}$) and evaporation rate coefficients ($\gamma$, s$^{-1}$)
of the $(FSA)_x(SA)_y(A)_z$ clusters were analyzed to estimate the thermodynamic stability of the
clusters involved in the SA-A-FSA system (Tables S6-S7). The $\Delta G$ and $\gamma$ of the important pure
SA·A clusters and FSA-containing stable clusters were primarily discussed at three temperature. At
298.15 K, the $\Delta G$ value of the SA·A cluster was 2.1 kcal·mol$^{-1}$ greater than that of the FSA·A cluster
(Fig. 5). Meanwhile, its $\gamma$ value was about 10 times greater than that of the FSA·A cluster, suggesting
that the FSA·A cluster is more stable and likely to participate in subsequent growth as an initial
cluster. For the $(FSA)_2·(A)_2$ cluster, its $\Delta G$ (-31.1 kcal·mol$^{-1}$) was smaller by 4.6 kcal·mol$^{-1}$ than that
of the $(SA)_2·(A)_2$ cluster (-26.5 kcal·mol$^{-1}$) with the $\gamma$ value of the former one ($5.34 \times 10^1$ s$^{-1}$) at least
$10^4$ times lower than that of the latter one ($6.13 \times 10^5$ s$^{-1}$), indicating that the $(FSA)_2·(A)_2$ cluster is
more stable than clusters containing SA and A with the same acid-base number. For the $(FSA)_3·(A)_3$
cluster, its $\gamma$ ($3.30 \times 10^{-1}$ s$^{-1}$) was nearly $10^3$ times lower than that of the $(SA)_3·(A)_3$ ($2.25 \times 10^2$ s$^{-1}$)
cluster, allowing $(FSA)_3·(A)_3$ to serve as a critical nucleation cluster and participate in subsequent
growth. Similarly, at 278.15 K and 258.15 K, the FSA·A, $(FSA)_2·(A)_2$ and $(FSA)_3·(A)_3$ clusters
were all more stable than the SA-A binary nucleation clusters with the same acid-base number.
Regarding for the $(FSA)_2·SA·(A)_3$·and FSA·$(SA)_2·(A)_3$ clusters at 298.15 K, the $\Delta G$ values (-56.7
and -54.1 kcal·mol$^{-1}$) were lower than that of $(SA)_3·(A)_3$ (-52.0 kcal·mol$^{-1}$). Simultaneously, the $\gamma$
values of the $(FSA)_2·SA·(A)_3$ ($8.49 \times 10^{-4}$ s$^{-1}$) and FSA·$(SA)_2·(A)_3$ ($5.75 \times 10^1$ s$^{-1}$) clusters were
respectively lower $10^6$ and 10 times lower than that of $(SA)_3·(A)_3$ ($2.25 \times 10^2$ s$^{-1}$). Likewise, the
$(FSA)_2·SA·(A)_3$ and FSA·$(SA)_2·(A)_3$ clusters were more stable than the $(SA)_3·(A)_3$ cluster at low
temperatures (278.15 K and 258.15 K) due to their significantly lower evaporation rates. Therefore,
compared to pure SA-A clusters, clusters containing FSA molecules exhibit higher stability and are
more likely to engage in nucleation and subsequent cluster growth processes as stable clusters. The
clusters of $(SA)_3·(A)_3$, $(FSA)_2·SA·(A)_3$ and FSA·$(SA)_2·(A)_3$·have the potential to further grow into
the boundary clusters [$(FSA)_2·(SA)_2·(A)_3$, $(FSA)_1·(SA)_3·(A)_3$ , $(SA)_4·(A)_3$ and $(SA)_4·(A)_4$,].

### 3.5 Influence of Particle Formation Rates Under Varying Temperatures and Nucleation Precursor Concentrations


To investigate the cluster formation rate ($J$, cm$^{-3}$·s$^{-1}$) and the enhancement factor ($R$) of cluster



formation rate by FSA, a range of ACDC simulations were performed using thermodynamic data
for the SA-A-FSA clusters at varying temperatures and monomer concentrations ([SA] = $10^4$ - $10^8$,
[A] = $10^7$ - $10^{11}$ and [FSA] = $10^3$ - $10^7$ molecules·cm$^{-3}$). The value of $R$ is defined as $R=J_{SA-A-FSA}/J_{SA-}$
$_A$. The values of $J$ and $R$ for the SA-A-FSA system at varying temperatures (Fig. 6) showed that $J$
increased as the temperature decreased, due to the smaller values of both $\Delta G$ and $\gamma$ at lower
temperatures. Meanwhile, $J$ increased with increasing [FSA], attributable to the formation of more
SA-A-FSA clusters. Variations in [FSA] and temperature can also affect $R$. A significant increase in
$R$ with the rising [FSA] has been observed, suggesting that FSA can strongly enhance the nucleation
rate in SA-A NPF. Interestingly, as the temperature increases (Fig. 6(b)), the value of $R$ becomes
greater. In summary, the inclusion of FSA can substantially improve $J$ for SA-A nucleation in
regions with relatively high [FSA] during summer or at lower altitudes with high temperatures.
In addition to temperature and [FSA], $J$ and $R$ can also be affected by [SA] and [A]. At 278.15
K, $J$ increased with increase of [SA] or [A] (Fig. 7(a)). Nevertheless, $R$ decreased with increasing
[SA] (Fig. 7(b)). This trend may be due to both FSA and SA are acidic molecules, creating a
competitive relationship when they interact with A. Additionally, the changes in $J$ with [SA] or [A]
and $R$ with [SA] were similar at other temperatures of 258.15 K and 298.15 K. Similar negative
dependencies between $R$ and [A] were observed at both 278.15 K and 298.15 K. This occurs because,
as the [A] increases, the interaction between FSA and SA in the SA-A-FSA system may be disrupted,
leading to a decrease in the saturation of FSA interaction sites and a reduction in $R$. Notably, at the
lower temperature of 258.15 K, when [FSA] was high, the value of $R$ initially decreased and then
increased with increasing [A] (as depicted in Fig. S12 (b)). This may be attributed to the following
reasons. First, as [A] increases, the interaction between FSA and SA in the ternary cluster may be
disrupted, leading to a decrease in the saturation of FSA interaction sites and a reduction in $R$. Then,
as the concentration of A further increases, excess A molecules bind to FSA molecules, leading to
an increase in $R$. In summary, FSA primarily enhances SA-A nucleation in regions with higher
temperatures and lower [A] and [SA].
**3.6 FSA-Driven Nucleation Enhancement Mechanism**
The clusters formed in the system via two main pathways: the pure SA-A pathway and SA-A-
FSA pathways (Fig. 8). The pure SA-A nucleation pathway primarily formed stable $(SA)_3·(A)_3$
clusters through monomer addition and collision with SA·A cluster. The SA-A-FSA nucleation



pathway can be categorized into two routes. One route involved the initial formation of the stable
cluster FSA·A, which then collided with one SA molecule, an FSA molecule, or another FSA·A
cluster to form subsequent stable clusters and continue growing. The other route involved the initial
formation of the stable $(SA)_2 \cdot A$ cluster, which then collided with one FSA·A cluster to form the
stable $(SA)_2 \cdot (A)_2 \cdot FSA$, continuing to grow through the addition of an A molecule or an FSA
molecule. Interestingly, at varying temperatures and concentrations of nucleating precursors, the
FSA molecule exhibited distinct effects and contributions in the SA-A system. As the temperature
increased, the contribution of the SA-A-FSA pathway rose from 51% to 97% (Fig. 9(a)). Therefore,
the cluster growth pathway involving FSA appears to prevail at relatively higher temperatures, such
as during summer or at lower altitudes. The involvement of FSA in the primary cluster formation
pathway may also be influenced by the concentration of the precursors. Specifically, the contribution
of the FSA participation pathway exhibited a negative correlation with [SA] or [A] at 278.15 K (Fig.
9(b-c)). Consequently, the contributions of the SA-A-FSA pathway may be more substantial in the
clean atmospheric boundary layer with low [A] and [SA], such as in area distant from heavy traffic
and emission sources of SA. Additionally, the contribution of the SA-A-FSA pathway increases as
[FSA] rises (Fig. 9(d)). At low [FSA] ($10^3$ molecules·cm$^{-3}$), the contribution of SA-A-FSA pathway
was only 35%, with cluster growth pathways predominantly governed by the formation of pure SA-
A clusters. However, as [FSA] increased to $10^4$ molecules·cm$^{-3}$, the contribution of FSA-involving
clusters rose to 84%, making the pathway involving FSA dominant for cluster formation in the SA-
A-FSA system. Moreover, the SA-A-FSA mechanism contributed more significantly (97%) at
higher [FSA] concentrations ($10^5$-$10^7$ molecules·cm$^{-3}$). In summary, consistent with the variation
observed in $R$ with temperature and precursor concentrations, the contribution of the pathway
involving FSA is significantly dominant in the NPF process with decreasing [SA] and [A] and
increasing temperature and [FSA]. These results suggest that FSA could be a significant contributor
to SA-A atmospheric NPF, and the SA-A-FSA pathway may dominate in regions with high FSA
emissions and relatively high temperatures.

## 387    **4. Summary and Conclusions**

The potential contribution of FSA to gaseous and interfacial $SO_3$ hydrolysis, as well as its
enhancement of atmospheric particle formation was investigated. Gaseous results indicated that $SO_3$



hydrolysis with FSA has a Gibbs free energy barrier as low as 1.5 kcal·mol$^{-1}$ and can effectively
compete with $SO_3$ hydrolysis by $HNO_3$ ($10^9$ molecules·cm$^{-3}$), HCOOH ($10^8$ molecules·cm$^{-3}$) and
$H_2SO_4$ ($10^6$ molecules·cm$^{-3}$) over a temperature range of 280.0-320.0 K. Interfacial BOMD
simulations illustrated that FSA-mediated $SO_3$ hydrolysis at the gas-liquid interface occurs through
a stepwise mechanism and can be completed within a few picoseconds. ACDC kinetic simulations
indicated that FSA significantly enhances cluster formation rates in the $H_2SO_4$-$NH_3$ system during
summer, increasing rates by more than $10^7$ times under conditions of high FSA concentrations and
low $H_2SO_4$ and $NH_3$ levels. The $H_2SO_4$-$NH_3$-FSA nucleation mechanism exhibits a stronger
nucleation ability than classical nucleation, making it a promising process for urban polluted
environments rich in FSA sources. Meanwhile, the interfacial species formed, such as $HSO_4^-$, $H_3O^+$
and $FSA^-$, act to attract precursor species (e.g., $H_2SO_4$, $NH_3$ and $HNO_3$) from the gas phase to the
nanodroplet interface, thereby facilitating further particle growth. This study broadens our
understanding of a novel $SO_3$ hydrolysis pathway involving FSA in polluted regions, identifies
previously overlooked new particle formation (NPF) sources in industrial areas, and deepens
knowledge of the atmospheric organic-sulfur cycle.
**Acknowledgments**
This work was supported by the National Natural Science Foundation of China (No: 22203052;
22073059; 42107109); the Key Cultivation Project of Shaanxi University of Technology (No:
SLG2101); the Education Department of Shaanxi Provincial Government (No. 23JC023).
**Declaration of competing interest**
The authors declare that they have no known competing financial interests or personal
relationships that could have appeared to influence the work reported in this paper.

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





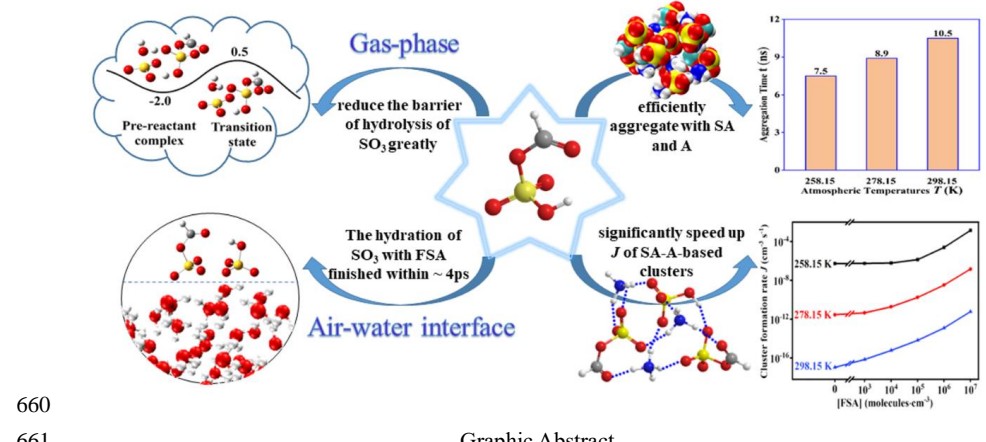


661                                                        Graphic Abstract





# Figure Captions

**Fig. 1.** Energy diagrams for $SO_3$ hydrolysis with FSA at the CCSD(T)-F12/cc-pVDZ-F12//M06-2X/6-311++G($2df$,$2pd$) level.

**Fig. 2.** BOMD simulations of $HSO_4^- \cdots FSA^- \cdots H_3O^+$ ion pair formation from $SO_3$ hydrolysis with FSA at the air-water interface. (Top: Snapshot structures from BOMD simulations, showing the ion pair formation. Bottom: Time evolution of key bond distances S-O1, O5-H3, and O1-H2 during the induced mechanism.)

**Fig. 3.** ESP-mapped vdW surfaces of sulfuric acid (SA), ammonia (A) and formic sulfuric anhydride (FSA). Blue, red, yellow, cyan, and white spheres represent N, O, S, C, and H atoms, respectively, with ESP in kcal·mol$^{-1}$.

**Fig. 4.** (a) Bar graph of aggregation completion time at different atmospheric temperatures; (b) snapshots of nucleation simulation at 258.15 K from FSA, SA and A using the VDW representation, with $N_2$ and $O_2$ shown using the line drawing method.

**Fig. 5.** Histogram of (a) Gibbs free energy of formation ($\Delta G$, kcal·mol$^{-1}$) and (b) evaporation rate coefficient ($\gamma$, s$^{-1}$) for key pure SA-A clusters and FSA-containing stable clusters at 258.15, 278.15 and 298.15 K.

**Fig. 6.** (a) Cluster formation rate ($J$, cm$^{-3}$ s$^{-1}$) and (b) enhancement factor $R$ with [SA] = $10^6$ molecules·cm$^{-3}$, [A] = $10^9$ molecules·cm$^{-3}$ at three temperatures (black: 258.15 K, red: 278.15 K, blue: 298.15 K).

**Fig. 7.** (a) The cluster formation rate ($J$, cm$^{-3}$ s$^{-1}$) and (b) enhancement factor $R$ as a function of [A] with [FSA] = $10^6$ molecules·cm$^{-3}$ at 278.15 K for five [SA] levels (black: $10^4$, red: $10^5$, blue: $10^6$, green: $10^7$, purple: $10^8$ molecules·cm$^{-3}$).

**Fig. 8.** Primary growth pathways of clusters at $T$ = 278.15 K, [SA] = $10^6$ molecules·cm$^{-3}$, [A] = $10^9$ molecules·cm$^{-3}$, and [FSA] = $10^3$-$10^7$ molecules·cm$^{-3}$. Blue and orange arrows represent the SA-A-based and SA-A-FSA-based pathways, respectively.

**Fig. 9.** Influence of (a) temperature, (b) [SA], (c) [A] and (d) [FSA] on the relative contribution of the pure SA-A pathway and the FSA-containing pathway to the flux out of the system.



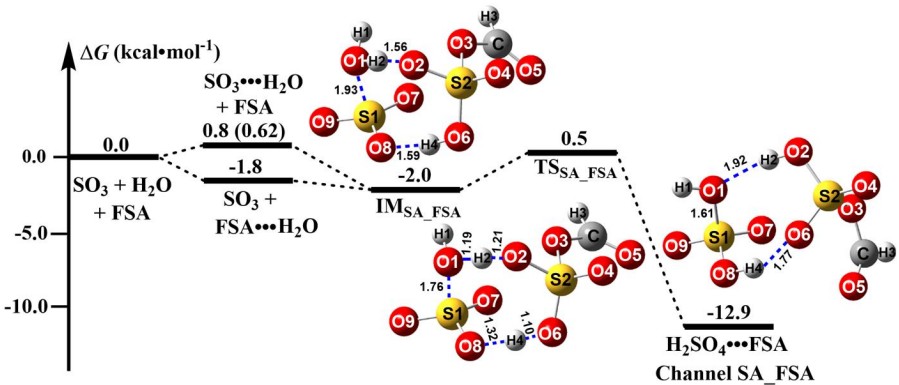


**Fig. 1.** Energy diagrams for $SO_3$ hydrolysis with FSA at the CCSD(T)-F12/cc-pVDZ-F12//M06-
2X/6-311++G(2$df$,2$pd$) level.





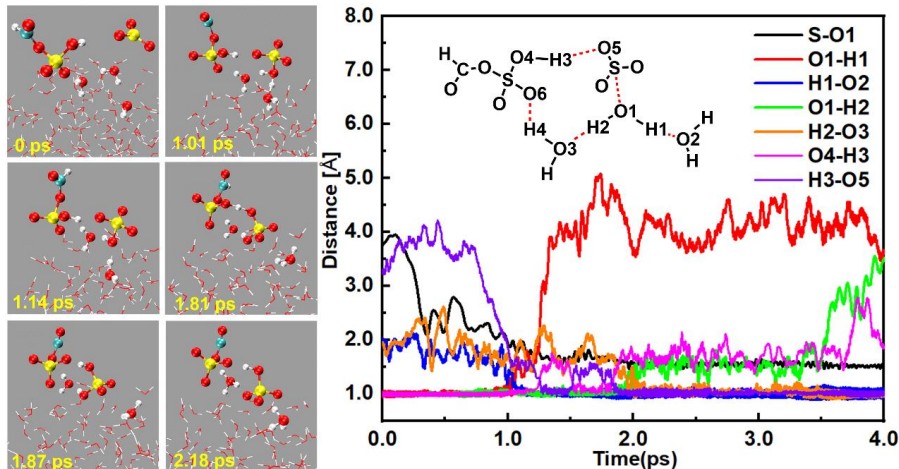


**Fig. 2.** BOMD simulations of $HSO_4^-$•••$FSA^-$•••$H_3O^+$ ion pair formation from $SO_3$ hydrolysis with

FSA at the air-water interface. (Top: Snapshot structures from BOMD simulations, showing the ion

pair formation. Bottom: Time evolution of key bond distances S-O1, O5-H3, and O1-H2 during the

induced mechanism.)




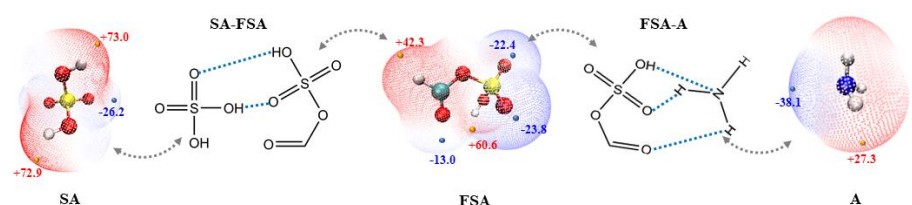


**Fig. 3.** ESP-mapped vdW surfaces of sulfuric acid (SA), ammonia (A) and formic sulfuric anhydride
(FSA). Blue, red, yellow, cyan, and white spheres represent N, O, S, C, and H atoms, respectively,
with ESP in kcal·mol$^{-1}$.



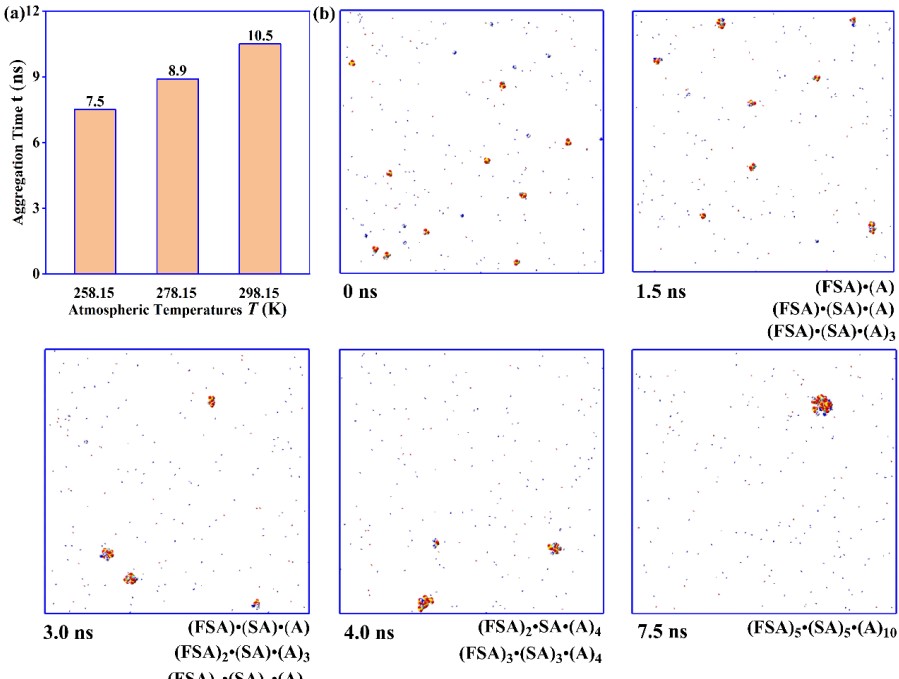


**Fig. 4.** (a) Bar graph of aggregation completion time at different atmospheric temperatures; (b)
snapshots of nucleation simulation at 258.15 K from FSA, SA and A using the VDW representation,
with $N_2$ and $O_2$ shown using the line drawing method.

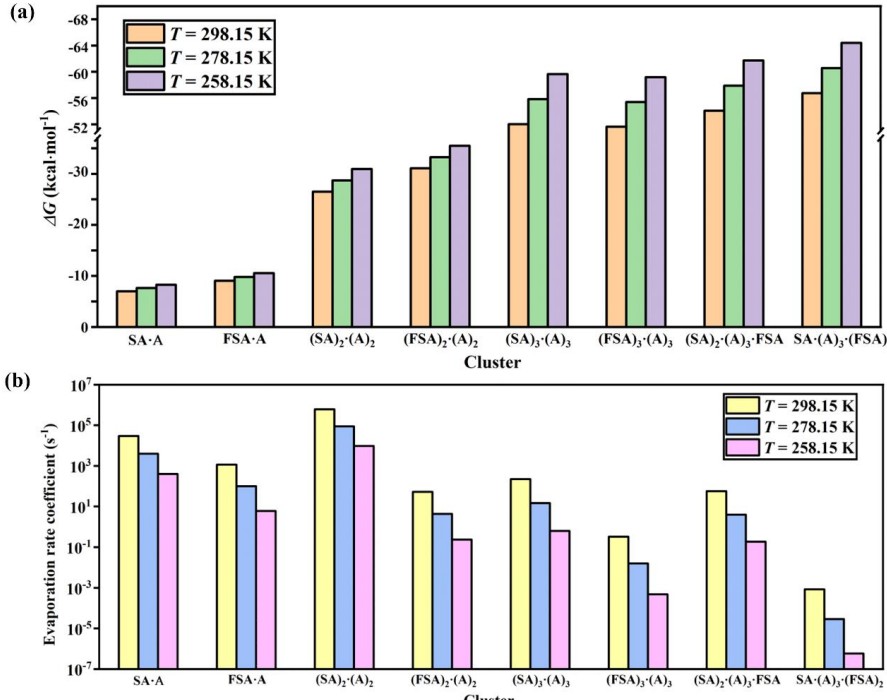

**Fig. 5.** Histogram of (a) Gibbs free energy of formation ($\Delta G$, kcal·mol$^{-1}$) and (b) evaporation rate coefficient ($\gamma$, s$^{-1}$) for key pure SA-A clusters and FSA-containing stable clusters at 258.15, 278.15 and 298.15 K.





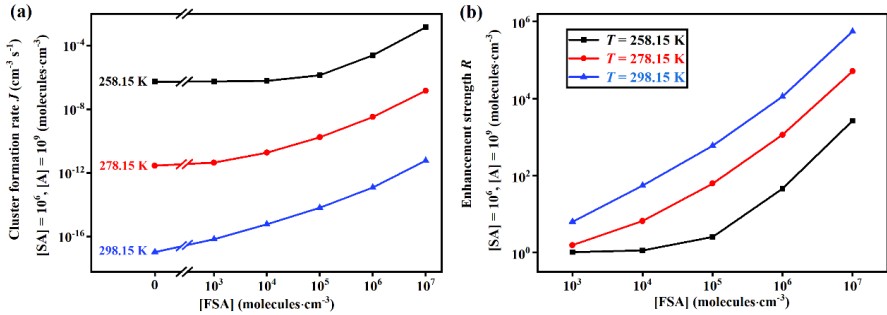

**Fig. 6.** (a) Cluster formation rate ($J$, cm$^{-3}$ s$^{-1}$) and (b) enhancement factor $R$ with $[SA] = 10^6$ molecules·cm$^{-3}$, $[A] = 10^9$ molecules·cm$^{-3}$ at three temperatures (black: 258.15 K, red: 278.15 K, blue: 298.15 K).



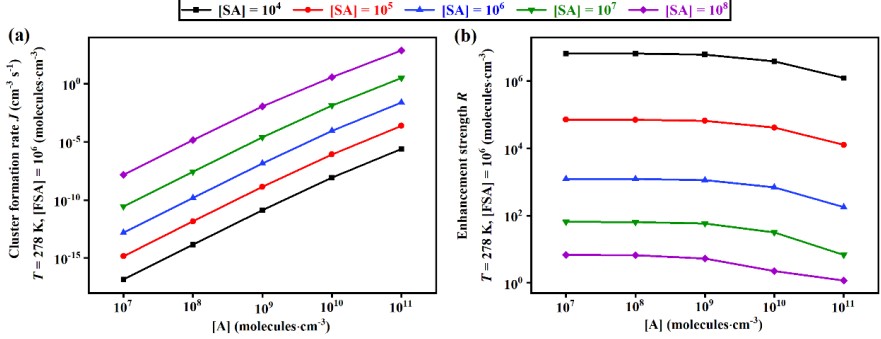

**Fig. 7.** (a) The cluster formation rate ($J$, cm$^{-3}$ s$^{-1}$) and (b) enhancement factor $R$ as a function of [A] with [FSA] = 10$^6$ molecules·cm$^{-3}$ at 278.15 K for five [SA] levels (black: 10$^4$, red: 10$^5$, blue: 10$^6$, green: 10$^7$, purple: 10$^8$ molecules·cm$^{-3}$).



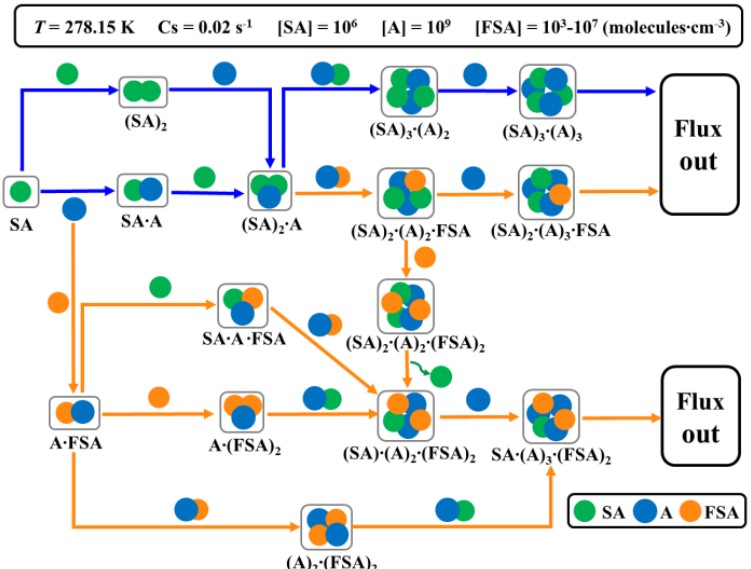

**Fig. 8.** Primary growth pathways of clusters at $T$ = 278.15 K, [SA] = $10^6$ molecules·cm$^{-3}$, [A] = $10^9$
molecules·cm$^{-3}$, and [FSA] = $10^3$-$10^7$ molecules·cm$^{-3}$. Blue and orange arrows represent the SA-A-
based and SA-A-FSA-based pathways, respectively.





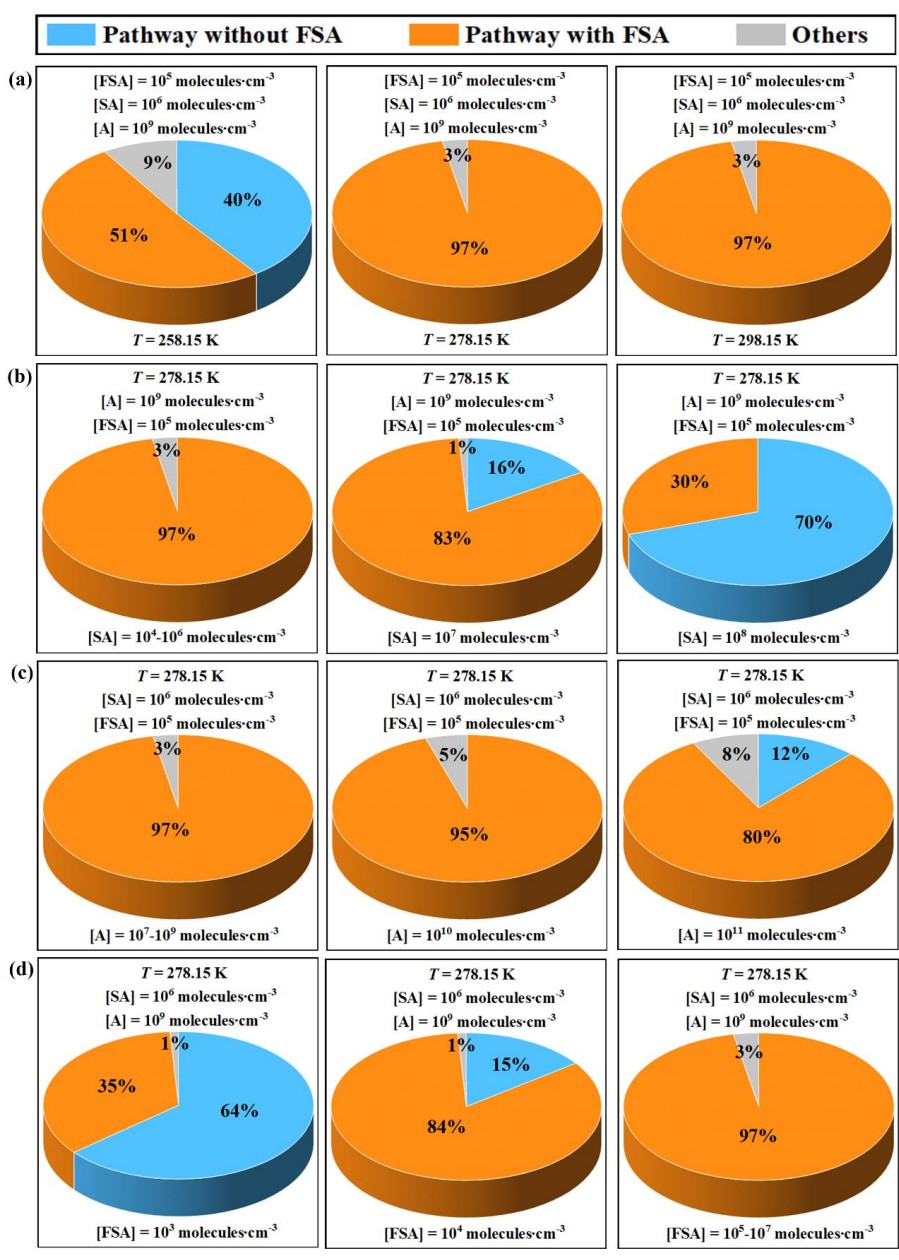

**Fig. 9.** Influence of (a) temperature, (b) [SA], (c) [A] and (d) [FSA] on the relative contribution of the pure SA-A pathway and the FSA-containing pathway to the flux out of the system.



**Table 1.** Rate constants ($cm^3 \cdot molecule^{-1} \cdot s^{-1}$) for $SO_3$ hydrolysis with and without FSA, $H_2O$, and $X$
($X$ = $HNO_3$, $HCOOH$, $(COOH)_2$ and $H_2SO_4$) within the temperature range of 280-320 K

| $T$/K | $k_{SA\_FSA}$ | $k_{SA}$ | $k_{SA\_WM}$ | $k_{SA\_FA}$ | $k_{SA\_NA}$ | $k_{SA\_OA}$ | $k_{SA\_SA}$ |
|---|---|---|---|---|---|---|---|
| 280 | $7.94 \times 10^{-11}$ | $6.24 \times 10^{-24}$ | $1.68 \times 10^{-12}$ | $8.88 \times 10^{-11}$ | $1.26 \times 10^{-12}$ | $8.02 \times 10^{-11}$ | $5.60 \times 10^{-11}$ |
| 290 | $7.84 \times 10^{-11}$ | $8.12 \times 10^{-24}$ | $1.45 \times 10^{-12}$ | $8.17 \times 10^{-11}$ | $1.05 \times 10^{-12}$ | $7.74 \times 10^{-11}$ | $5.08 \times 10^{-11}$ |
| 298 | $7.71 \times 10^{-11}$ | $1.02 \times 10^{-23}$ | $1.28 \times 10^{-12}$ | $7.60 \times 10^{-11}$ | $9.11 \times 10^{-13}$ | $7.48 \times 10^{-11}$ | $4.69 \times 10^{-11}$ |
| 300 | $7.67 \times 10^{-11}$ | $1.09 \times 10^{-23}$ | $1.24 \times 10^{-12}$ | $7.46 \times 10^{-11}$ | $8.80 \times 10^{-13}$ | $7.42 \times 10^{-11}$ | $4.59 \times 10^{-11}$ |
| 310 | $7.46 \times 10^{-11}$ | $1.50 \times 10^{-23}$ | $1.07 \times 10^{-12}$ | $6.78 \times 10^{-11}$ | $7.46 \times 10^{-13}$ | $7.06 \times 10^{-11}$ | $4.13 \times 10^{-11}$ |
| 320 | $7.21 \times 10^{-11}$ | $2.12 \times 10^{-23}$ | $9.22 \times 10^{-13}$ | $6.12 \times 10^{-11}$ | $6.46 \times 10^{-13}$ | $6.68 \times 10^{-11}$ | $3.70 \times 10^{-11}$ |




**Table 2.** Binding free energy (kcal·mol$^{-1}$) for the formation of various clusters at 298 K.

| FSA$^-$-SA | FSA$^-$-HNO$_3$ | H$_3$O$^+$-A | H$_3$O$^+$-SA | SA-A |
|---|---|---|---|---|
| -21.2 | -12.1 | -51.7 (-49.2)[a] | -27.5 (-27.0)[a] | -8.9 (-8.9)[a] |
| HSO$_4^-$-SA | HSO$_4^-$-(COOH)$_2$ | HSO$_4^-$-HNO$_3$ | SA-A-FSA$^-$ | SA-A-HOOCCH$_2$COOH |
| -41.6 | -33.6 | -27.8 | -25.6 | -13.1(13.6)[b] |
| SA-A-HOCCOOSO$_3$H | SA-A-CH$_3$OSO$_3$H | SA-A-HOCH$_2$COOH | SA-A-HOOCCH$_2$CH(NH$_2$)COOH | |
| -20.4 (-22.5)[c] | -18.8 (-20.7)[d] | -13.2 (-14.0)[e] | -12.8 (-13.5)[f] | |

Energies are given in kcal·mol$^{-1}$ and calculated at the M06-2X/6-311++G(2*df*,2*pd*) level of theory. References are as
follows: [a] Zhong et al. (2019), [b] Zhang et al. (2018), [c] Rong et al. (2020), [d] Gao et al. (2023), [e] J. Liu et al. (2021), [f]
Zhang et al. (2017).