# Peer review of "Enhancing SO3 Hydrolysis and Nucleation: The Role of"

_EGUsphere, 2024_

## Referee Comment (RC2)

Rui Wang and co-workers studied the enhancing effect of FSA on $SO_3$ hydrolysis, both in the gas phase and at the gas-liquid nanodroplet interface, as well as its impact on $H_2SO_4$-$NH_3$-driven NPF through quantum chemical calculations, atmospheric clusters dynamics code (ACDC) kinetics combined with Born-Oppenheimer molecular dynamics (BOMD). The present study identified a novel $SO_3$ hydrolysis pathway involving FSA in the polluted regions and FSA enhanced nucleation mechanism.

The work is quite comprehensive and highly routine in nature which involves substantial workload. However, there are some contradictory points/conclusions that may make the readership confused. Anyway, I hope the authors find the below comments useful.

**Line 103:** "Initially, the ABCluster program was utilized to randomly produce n $\times$ 1000 initial isomers (where n = 2 to 4)..."
**Q1**: Why not adopt a unified "n" value for all the clusters ?
**Q2**: The M06-2X functional with a 6-311++G(2df,2pd) basis set is a reasonable computational level of theory for studying kinetics. However, it would be appreciable if the authors could further motivate their choice. There certainly exist numerous benchmarks of the performance of different DFT functionals for thermochemistry and barrier heights of atmospheric relevant systems.
**Q3**: CCSD(T)-F12/cc-pVDZ-F12 was used for the single point calculations. Please also specify which basis sets were used for the resolution of identity (RI) and complete auxiliary basis set (CABS) parts?
**Q4**: To generate more accurate gibbs free energies, the authors calculated the FSA catalytic $SO_3$ hydrolysis reaction mechanism under the CCSD(T)-F12/cc-pVDZ-F12//M06-2X/6-311++G(2df,2pd) level of theory. However, the calculations for the nucleation clusters were just under the M06-2X/6-311++G(2df,2pd) level of theory. No further single point calculations were performed, even the CCSD(T)-F12/cc-pVDZ-F12 calculations in ORCA are quite fast. I suggest the authors insist on using CCSD(T)-F12/cc-pVDZ-F12//M06-2X/6-311++G(2df,2pd) in the whole calculations, including the reactions and the nucleation process, the consistency in the calculation method throughout the study will also make it more scientific and reasonable.

**2.2 Rate Coefficient Computations**
The studied reactions involves the motion of a hydrogen atom may have a high imaginary frequency. So the tunneling effects could be quite profound. Has the tunneling correction been considered in the rate coefficient computations ? If so, please clarify in section 2.2. If not, I highly recommend taking into account the tunneling correction.

**2.3 BOMD Simulations**
**Q1:** Line 126: Please specify the detailed basis set used by the BOMD simulations.
**Q2:** Line 130 "... to eliminate periodic boundary conditions with step of 0.5 fs..."

Line 135 "...neighbouring water droplets, using a step of 1.0 fs..."
In order to ensure the readability of the manuscript, please briefly explain why two different time steps were used here?

**2.4 Molecular Dynamics Simulation of Nucleation**

**Q1:** Line 141 "M06-2X/6-311++G(*2df,2pd*)" → M06-2X/6-311++G(2df,2pd). There's no need for italics here. Please also check the whole manuscript.

**Q2:** Line 142 "Multiwfn 3.8" Actually, there is no Multiwfn 3.8 version. Please check the Multiwfn website and ensure the exact version which the authors downloaded and used, maybe it is Multiwfn 3.8 (dev).

**Q3:** Line 151 "Bond lengths were restricted by the LINCS algorithm". Were all the bond lengths in the MD simulations restricted by the LINCS algorithm?

**2.5 Atmospheric Cluster Dynamics Code (ACDC) Model**

**Q1:** Lines 160-161 "...while $\beta_{i,j}$ stands for the collision rate between $i$ and $j$ clusters. The term $\gamma_{(i+j)\to i}$ ...". Actually, $\beta_{i,j}$ here is not collision rate but collision coefficient,$\gamma$ is not evaporation rate but evaporation coefficient. The present misleading expression may confuse readership in their understanding of the birth-death equation. Please revise.

**Q2:** Line 167 "...were selected as the boundary clusters in the SA-A-FSA system".
Whether a cluster should be selected as a boundary cluster is related to both its evaporation rate and collision rate, as referenced in the article on the ACDC model. Please clarify the selection criteria for boundary clusters in the present study. Furthermore, in lines 313-315, the author pointed out that the evaporation coefficient of $(FSA)_3(A)_3$ ($3.30\times10^{-1}$ $s^{-1}$) was nearly $10^3$ times lower than that of $(SA)_3(A)_3$ ($2.25\times10^2$ $s^{-1}$). Then why clusters $(FSA)_4(A)_3$, $(FSA)_4(A)_4$, $(FSA)_3(SA)(A)_3$ were not considered as the boundary clusters ? This is the first contradictory point.

**Lines 252-255** "In contrast to the $SO_3$ hydrolysis with FSA in the gas phase, which does not proceed within 100 ps, the reaction at the gas-liquid nanodroplet interface rapidly proceeds within just a few picoseconds. This indicates that interfacial water molecules at the gas-liquid nanodroplet interface can accelerate the SO3 hydrolysis."

**Q1:** Since the FSA catalytic $SO_3$ hydrolysis reaction on the gas-liquid nanodroplet interface is faster, it is more likely to occur on the interface in the case of the low saturated vapor pressure of FSA, making the FSA catalytic reaction in the gas phase relatively less important. And the generated SA molecules at the nanodroplet interface tends to deprotonate rather than evaporate into the gas phase. This is the second contradictory point. Is it possible to calculate the proportion of FSA catalytic reaction occurring in the gas phase versus on the interface? If the proportion of gas phase reaction only accounts for a small fraction, then emphasizing the role of FSA in gas phase reaction and nucleation would appear to have limited significance.

**Q2**: Since different molecules in the gas phase can catalyze the $SO_3$ hydrolysis reaction, and water molecules exhibit the fastest catalytic rate due to their higher concentration in the gas phase. So it is necessary to supplement the calculation on the

catalytic effect of water molecules on the interface, where their concentration is even higher. A comparison should hence be made between the catalytic effect of water molecules and FSA molecule on the interface.

**Lines 261-264**: "As shown, the interactions of FSA$^-$-SA (-21.2 kcal $\cdot$ mol$^{-1}$) and FSA$^-$-HNO3 (- 12.1 kcal $\cdot$ mol$^{-1}$) are stronger than that of SA-A (-8.9 kcal $\cdot$ mol$^{-1}$), illustrating that interfacial FSA$^-$ and H3O$^+$ ions can attract precursor molecules from the gaseous phase to the aqueous nanodroplet surface, and thus facilitating particle growth."

**Q:** Since the reaction rate for the FSA catalytic SO$_3$ hydrolysis on the aqueous nanodroplet surface is much higher than that in the gas phase. And the generated SA molecules at the nanodroplet interface tends to deprotonate rather than evaporate into the gas phase. Furthermore, interfacial FSA$^-$ ions can also attract precursor molecules from the gas phase to the aqueous nanodroplet surface, and thus facilitate particle growth. Combining with the low saturated vapor pressure of FSA, it seems that FSA is more likely to contribute to the particle growth. In other words, is that possible to calculate the contribution proportion of FSA to the gas phase nucleation and the particle growth ? If the contribution proportion of FSA to the gas phase nucleation only accounts for a small fraction, then the FSA-involved nucleation in Section 3.3 and 3.4 would appear to have limited significance. This is the third contradictory point.

**Line 266:** SA-A-Y (Y=HOOCCH2COOH, HOCCOOSO3H, CH3OSO3H, HOOCCH2CH(NH2)COOH and HOCH2COOH) clusters.

**Q:** Such comparison does not seem that fair and reasonable. Y and FSA are all acid molecules which can be deprotonated at the nanodroplet interface. So it is more reasonable to compare the binding free energies of SA-A-FSA$^-$ and SA-A-Y$^-$ .

**3.3 FSA's Role in Nucleation and Cluster Formation**

**Q1:** Line 283 "In these simulation systems, 5 FSA, 5 SA, 10 A, 20 H2O, 41 O2 and 154 N2 molecules were included."

The ratio of the number of molecules does not seem to be consistent with the real atmospheric concentration condition. Once the number of molecules in the Gromacs MD simulation is large enough, molecular aggregation will take place.

**Q2: Fig.4.**

The classical MD simulation performed by Gromacs relies on the force field of the molecules, which can only describe physical aggregation. However, during the nucleation process, the proton transfer between acid and base molecules plays an important role in acid-base nucleation which can not be reflected in the classical MD simulation. In this case, the physical aggregation pathways in the MD simulations are not completely consistent with those in ACDC simulations. For example, in the MD simulation shown in Fig. 4, all the clusters containing SA, A and FSA. In the ACDC simulations, there exist pure SA-A paths. The clusters growth pathways are not completely consistent with each other. This is the fourth contradiction in this

==manuscript== which may make the readership confused.

**Line 352:** "First, as [A] increases, the interaction between FSA and SA in the ternary cluster may be disrupted, leading to a decrease in the saturation of FSA interaction sites and a reduction in R."

**Q:** According to the data shown in Fig.5, the $\Delta G$ and evaporation coefficient of FSA-based clusters is much lower than that of SA-based clusters. As [A] increases, if the interaction between FSA and SA in the ternary cluster was disrupted, the FSA will interact with the increasing A. The generated FSA-A-based clusters are more thermodynamically stable, and hence the R will increase. This is somewhat inconsistent with the current conclusion.

**Line 386**: "...with high FSA emissions..."
**Q:** Is there any emissions sources of FSA?

**Line 592**: Smith, C. J., Huff, A. K., Ward, R. M., and Leopold, K. R.: Carboxylic sulfuric anhydrides, J. Phys. Chem. A, 124, 601-612, https://doi.org/10.1126/science.1180315, 2020.
**Q:** The URL (https://doi.org/10.1126/science.1180315) provided does not match this reference, the author wrote a wrong URL. Please revise and also check the whole references section.

**Technical corrections:**
(1) Line 72: A full stop "." should be added at the end of the paragraph.
(2) Line 102: "The most stable structure of the (FSA)x(SA)y(A)z (z $\leqslant$ x + y $\leqslant$ 3)...". This is the first time that abbreviation "A" appeared. It will be more clear to clarify that "A" is the abbreviation of $NH_3$.
(3) Results and discussion: The tense of the sentences throughout the "Results and discussion" should be consistent. If the past tense is used, please use the past tense uniformly. For example, in lines 173 and 176, the authors used "was" and "is", respectively.
(4) Line 376: "low"→"lower". line 386: "high"→ "higher".
(5) Fig.4: "VDW"→"vdW".
(6) Fig.7: "$T$=278 K"→"$T$=278.15 K"

---

## Author Response (AR2)

**Responses to Referee #1's comments**

We are grateful to the reviewers for their valuable and helpful comments on our manuscript

"Enhancing $SO_3$ Hydrolysis and Nucleation: The Role of Formic Sulfuric Anhydride" (MS No.:

**egusphere-2024-3275**). We have revised the manuscript carefully according to reviewers'

comments. The point-to-point responses to the Referee #1's comments are summarized below:

**General comments:**

This manuscript reports that the formic sulphuric anhydride molecule (FSA) is able to catalyze the hydration of $SO_3$ to form sulphuric acid in the atmosphere. This should not be a surprise to anyone in this field, since it seems that any molecule which is able to form a complex with $SO_3$ and $H_2O$

and help the double proton transfer happen will catalyze this process. It was still worth doing this work to get some quantitative data for this new system, though, and it should be published with some minor revisions.

**Specific Comments**:

**Comment 1.**

I have some concerns about the simulations done to demonstrate the propensity of the pre-reactive cluster to sit on the interface (results shown in Fig S6). I was glad to see this done, since it is often omitted from similar work, but it is very important to do; who cares if the reaction is faster at the interface, if the pre-reactive complex never resides there?

However, I could not find any discussion of the simulation methods used. I assume that these were not ab initio simulations, since they were quite long (150ns). But the only force-field based simulations described in the manuscript were the clustering simulations described in section 2.4.

Simulating the interfacial preference of different species is known to be difficult, it is sensitive to, for example, whether polarizability is included in the force field, as well as the analysis methods used to define the interface (see eg. https://doi.org/10.1080/08927022.2021.1980215 for a recent review) so it is crucial that these details be included.

**Response:** Thanks for your valuable comments. As stated by the reviewer, the surface preference of the $SO_3$-FSA complex was studied using classical molecular dynamics (MD) simulations rather than ab initio simulations. The corresponding revision has been respectively made as follows.

(a) The MD simulations were performed using GROMACS 2024.3 software package with the general AMBER force field (GAFF). GAFF is a complete force field; it covers almost all the organic chemical space that is made up of C, N, O, S, P, H, F, Cl, Br and I. The GAFF force field has been widely used in studies involving the air-water interface, demonstrating its suitability for predicting the properties of species at air-water interface. So, in Lines 144-155 on Page 5-6 of the revised manuscript, the sentence of "MD simulations were conducted using the GROMACS 2024.3 software package (Abraham et al., 2024) with the general AMBER force field (GAFF). GAFF is a comprehensive force field that encompasses nearly all of organic chemical space, including elements such as C, N, O, S, P, H, F, Cl, Br, and I. This force field has been widely utilized in studies of the air-water interface, with the results confirming its suitability for predicting the properties of species at this interface (Li et al., 2024b; Cheng et al., 2025; Zhao et al., 2019). To get the force field parameters, geometry optimization at the M06-2X/6-311++G(2df,2pd) level were performed, following Electrostatic potential (ESP) calculations at the same level. Geometry optimization and electrostatic potential (ESP) calculations were carried out with the Gaussian 09 software. The restrained electrostatic potential (RESP) charges were calculated using Multiwfn 3.8 (dev) (Lu and Chen, 2012). Subsequently, the AMBER parameter and coordinate files were generated using Packmol (Martínez et al., 2009) and Sobtop (Lu, 2023), respectively." has been reorganized.

(b) A cubic box of 4 nm side length with 2165 water molecules was firstly built. The water box was extended to 9 nm along the z axis direction, we put the water slab in the middle of the box with the COM coordinate of (2.0 nm, 2.0 nm, 4.5 nm) and a $SO_3$-FSA complex at (2.0 nm, 2.0 nm, 7.5 nm) (Fig. S6(c)). A 150 ns NVT simulation was performed. So, in the section of "2.4.1 Surface preference of $SO_3$, FSA and $SO_3$-FSA" of the revised manuscript, in Lines 157-161 on Page 6, the sentence of "A cubic box with a side length of 4 nm, containing 2165 water molecules, was initially constructed. The box was then extended along the z-axis to a length of 9 nm. The water slab was positioned at the center of the box with the COM coordinates of (2.0 nm, 2.0 nm, 4.5 nm), while the $SO_3$, FSA and $SO_3$-FSA complexes were placed at (2.0 nm, 2.0 nm, 7.5 nm) (Fig. S6(c)). Subsequently, a 150 ns NVT simulation was conducted." has been added.

  **Responses to Referee #2's comments**

We are grateful to the reviewers for their valuable and helpful comments on our manuscript

"Enhancing $SO_3$ Hydrolysis and Nucleation: The Role of Formic Sulfuric Anhydride" (MS No.:

**egusphere-2024-3275**). We have revised the manuscript carefully according to reviewers'

comments. The point-to-point responses to the Referee #2's comments are summarized below:

**Referee Comments:**

Rui Wang and co-workers studied the enhancing effect of FSA on $SO_3$ hydrolysis, both in the gas phase and at the gas-liquid nanodroplet interface, as well as its impact on $H_2SO_4$-$NH_3$-driven

NPF through quantum chemical calculations, atmospheric clusters dynamics code (ACDC) kinetics combined with Born-Oppenheimer molecular dynamics (BOMD). The present study identified a novel $SO_3$ hydrolysis pathway involving FSA in the polluted regions and FSA enhanced nucleation mechanism.

The work is quite comprehensive and highly routine in nature which involves substantial workload. However, there are some contradictory points/conclusions that may make the readership confused. Anyway, I hope the authors find the below comments useful.

**Response:** We would like to thank the reviewer for the positive and valuable comments, and we have revised our manuscript accordingly.

**Specific Comments**:

**Comment 1.**

Line 103: "Initially, the ABCluster program was utilized to randomly produce n × 1000 initial isomers (where n = 2 to 4)..."

Why not adopt a unified "n" value for all the clusters?

**Response:** Thanks for your valuable comments. Indeed, a multi-path searching approach is utilized to explore the stable structures of $(FSA)_x(SA)_y(A)_z$ ($z \leq x + y \leq 3$). For each global minimum cluster of $(FSA)_x(SA)_y(A)_z$ ($z \leq x + y \leq 3$), $n$ different searching pathways are considered.

Additionally, a single monomer is incorporated to form a larger cluster on top of the existing smaller ones. For instance, in the process of searching for the stable structure of $(SA)_2 \cdot (A)$ clusters, two search pathways exist: $(SA) \cdot (A) + SA$ and $(SA)_2 + A$. Similarly, in the search for the stable structure of (FSA)·(SA)·(A) clusters, three pathways are considered: (SA)·(A) + FSA, (SA)·(FSA) + A and (FSA)·(A) + SA. Additionally, we apologize for the incorrect range of $n$ values previously used. Upon reviewing all the search pathways, we confirm that the correct range for $n$ is $1 \leq n \leq 3$, rather than n = 2 to 4. Consequently, the sentence of "Initially, the ABCluster program was utilized to randomly produce $n \times 1000$ initial isomers (where $n$ = 2 to 4), which were subsequently evaluated using the PM6 method via MOPAC 2016 (Partanen et al., 2016)." has been changed as "Initially, the ABCluster program was utilized to randomly produce $n \times 1000$ initial isomers ($1 \leq n \leq 3$), which were subsequently evaluated using the PM6 method via MOPAC 2016 (Partanen et al., 2016)." in Lines 106-108 on Page 4 of the revised manuscript.

**Comment 2.**

The M06-2X functional with a 6-311++G(2df,2pd) basis set is a reasonable computational level of theory for studying kinetics. However, it would be appreciable if the authors could further motivate their choice. There certainly exist numerous benchmarks of the performance of different DFT functionals for thermochemistry and barrier heights of atmospheric relevant systems.

**Response:** Thanks for your valuable comments. According to the reviewer's suggestion, the M06-2X method with the 6-311++G(2df,2pd) basis set has been selected for the following reasons.

(a) It has been demonstrated that M06-2X functional is among the most effective functionals for describing noncovalent interactions and for estimating the thermochemistry and equilibrium structures of atmospheric reactions. Accordingly, the sentence of "The M06-2X functional (Mardirossian and Head, 2016; Pereira et al., 2017) is highly effective in describing noncovalent interactions and estimating the thermochemistry and equilibrium structures of atmospheric reactions." has been added in Lines 92-94 on Page 4 of the revised manuscript.

(b) The geometric parameters of the $SO_3$ and HCOOH reactants, calculated at the M06-2X/6-311++G(2df,2pd) level, are presented in Fig. S1. As seen in Fig. S1, the mean absolute deviations between the calculated bond distances and bond angles at the M06-2X/6-311++G(2df,2pd) level and the experimental data are 0.01 Å and 0.60°, respectively. This reveals that the calculated bond distances and bond angles at the M06-2X/6-311++G(2df,2pd) level are consistent with the available experimental data (From the NIST chemistry webbook, http://webbook.nist.gov/chemistry.).

Besides, the bond lengths and angles obtained from the M06-2X/6-311++G(2df,2pd) level are close to the values calculated at the M06-2X/6-311++G(3df,3pd) level (Fig. S1). Thus, the 6-

311++G(2df,2pd) basis set was selected for all M06-2X calculations, as it provides an optimal balance between accuracy and computational efficiency when compared to the 6-311++G(3df,3pd).

The corresponding revision has been shown in Fig. S1. Thus, in Lines 98-100 on Page 4 of the revised manuscript, the sentence of "It is noted that the calculated bond distances and bond angles at the M06-2X/6-311++G(2df,2pd) level (Fig. S1) are in good agreement with both experimental data and values obtained using the M06-2X/6-311++G(3df,3pd) method." has been added.

[Figure]

**Fig. S1** The optimized geometries of $SO_3$ and HCOOH, especially the main bond lengths and bond angles at two different theoretical levels. [a] The values obtained at the M06-2X/6-311++G(2df,2pd)

level of theory. [b] The values obtained at the M06-2X/6-311++G(3df,3pd) level of theory. [c] The values in parentheses are the experimental values. Bond length is in angstrom and angle is in degree.

**Comment 3.**

CCSD(T)-F12/cc-pVDZ-F12 was used for the single point calculations. Please also specify which basis sets were used for the resolution of identity (RI) and complete auxiliary basis set (CABS) parts?

**Response:** Thanks for your valuable comments. We apologize for the omission of the specific

CCSD(T)-F12/cc-pVDZ-F12 basis set used in the single-point energy calculations. So, the specific basis sets has been provided in Lines 102-104 on Page 4 of the revised manuscript, which has been organized as "To enhance the reliability of the relative Gibbs free energies, single-point energies at the CCSD(T)-F12/cc-pVDZ-F12-CABS level were calculated using the ORCA software (Neese,

2012)."

**Comment 4**

To generate more accurate gibbs free energies, the authors calculated the FSA catalytic $SO_3$

hydrolysis reaction mechanism under the CCSD(T)-F12/cc-pVDZ-F12//M06-2X/6-311++G(2df,2pd) level of theory. However, the calculations for the nucleation clusters were just under the M06-2X/6-311++G(2df,2pd) level of theory. No further single point calculations were performed, even the CCSD(T)-F12/cc-pVDZ-F12 calculations in ORCA are quite fast. I suggest the authors insist on using CCSD(T)-F12/cc-pVDZ-F12//M06-2X/6-311++G(2df,2pd) in the whole calculations, including the reactions and the nucleation process, the consistency in the calculation method throughout the study will also make it more scientific and reasonable.

**Response:** We sincerely thank the reviewer for the valuable comments. According to the reviewer's suggestion, single-point energies were recalculated for the optimized geometries of the stable $(FSA)_x(SA)_y(A)_z$ ($z \leq x + y \leq 3$) clusters, initially obtained at the M06-2X/6-311++G(2df,2pd) level, using the DLPNO-CCSD(T)-F12/cc-pVDZ-F12-CABS method in ORCA. The resulting changes based on the DLPNO-CCSD(T)-F12/cc-pVDZ-F12-CABS single-point energies are outlined below.

(a) In lines 111-113 Page 4 of the revised manuscript, the single point energies of the stable $(FSA)_x(SA)_y(A)_z$ ($z \leq x + y \leq 3$) clusters has been organized as "Lastly, based on the optimized geometries of the stable clusters at the M06-2X/6-311++G(2df,2pd) level, the single point energies were calculated at the DLPNO-CCSD(T)-F12/cc-pVDZ-F12-CABS level (Tchinda et al., 2022) using the ORCA.".

(b) Based on the energies at the DLPNO-CCSD(T)-F12/cc-pVDZ-F12-CABS//M06-2X/6-311++G(2df,2pd) level, Gibbs free energy of formation ($\Delta G$, kcal·mol$^{-1}$) and evaporation rate coefficient ($\gamma$, s$^{-1}$) of key pure SA-A clusters and FSA-containing stable clusters were reorganized in revised Fig. 5. Meanwhile, the corresponding Gibbs free energies of formation ($\Delta G$, kcal·mol$^{-1}$), evaporation rate coefficients ($\gamma$, s$^{-1}$), collision coefficients ($\beta$, cm$^3$ s$^{-1}$), total evaporation coefficients ($\sum\gamma$, s$^{-1}$) and ratios ($\beta C/\sum\gamma$) of monomer collisions for the $(FSA)_x(SA)_y(A)_z$ clusters has been recalculated in Table S6-S10 in the revised supporting information. In the whole revised manuscript, the values of Gibbs free energy of formation ($\Delta G$, kcal·mol$^{-1}$) and evaporation rate coefficient ($\gamma$, s$^{-1}$) for the $(FSA)_x(SA)_y(A)_z$ ($z \leq x + y \leq 3$) clusters have been updated.

[Figure]

**Fig. 5.** Histogram of (a) Gibbs free energy of formation ($\Delta G$, kcal·mol$^{-1}$) and (b) evaporation rate coefficient ($\gamma$, s$^{-1}$) for key pure SA-A clusters and FSA-containing stable clusters at 258.15, 278.15 and 298.15 K.

(c) Based on the Gibbs free energy of formation at the DLPNO-CCSD(T)-F12/cc-pVDZ-F12-CABS//M06-2X/6-311++G(2df,2pd) level, the cluster formation rate ($J$, cm$^{-3}$·s$^{-1}$), growth pathways and contribution of the SA-A-FSA clusters have been recalculated at varying temperatures and monomer concentrations, which has been reorganized in Fig. 6-9 in the revised manuscript.

**Comment 5.**

The studied reactions involves the motion of a hydrogen atom may have a high imaginary frequency. So the tunneling effects could be quite profound. Has the tunneling correction been considered in the rate coefficient computations? If so, please clarify in section 2.2. If not, I highly recommend taking into account the tunneling correction.

**Response:** Thanks for your valuable comments. In the MESMER program package, the Eckart potential function is commonly used to estimate quantum mechanical tunneling corrections to theoretically determined chemical rate coefficient calculations (***Nat. Commun.***, 2023, 14, 498; ***Phys.***

*Chem. Chem. Phys.,* 2023, 25, 28205-28212; *J. Phys. Chem. A,* 2019, 123, 8448-8459). In the present work, the Eckart tunneling correction was incorporated into the calculations of the rate coefficient for FSA-assisted $SO_3$ hydrolysis, with the Eckart tunneling correction included in the MESMER input file. We apologize for not mentioning that the Eckart tunneling correction in the previous version of the manuscript. Consequently, in Lines 124-125 on Page 5 of the revised manuscript, we have added the following statement: "Additionally, the MESMER calculations in this study applied an Eckart tunneling correction to the reaction rates."

**Comment 6.**

Line 126: Please specify the detailed basis set used by the BOMD simulations.

**Response:** Thanks for your valuable comments. In the BOMD simulations for FSA-assisted $SO_3$ hydrolysis in the gas phase and on a water droplet, the specific basis sets has been reorganized in Lines 131-133 on Page 5 of the revised manuscript and has been written as "The GTH norm-conserving pseudopotentials (Goedecker et al., 1996), along with the Gaussian DZVP basis set (Phillips et al., 2005) and the auxiliary plane wave basis set, were utilized to describe the core and valence electrons, respectively."

**Comment 7**

Line 130 "... to eliminate periodic boundary conditions with step of 0.5 fs..

Line 135 "...neighbouring water droplets, using a step of 1.0 fs..."

In order to ensure the readability of the manuscript, please briefly explain why two different time steps were used here?

**Response:** We thank you for your valuable comments and sincerely apologize for the reviewer's misunderstanding regarding the time steps. Consistent with previous studies (*Atmos. Environ.,* 2020, 230, 117514; *Atmosphere,* 2022, 14, 30; *Chemosphere*, 2020, 252, 126292), the time step in the gas-phase BOMD simulations was set to 0.5 fs. Similarly, as in previous studies (*Sci. Total Environ.,* 2024, 949, 174877; *Chemosphere*, 2021, 280, 130709; *Sci. Total Environ*., **2020**, 707, 135804), the time step for the BOMD simulations of the gas-liquid nanodroplet interface was set to 1.0 fs.

**Comment 8**

Line 141 "M06-2X/6-311++G(*2df,2pd*)"→M06-2X/6-311++G(2df,2pd). There's no need for italics here. Please also check the whole manuscript?

**Response:** Thanks for your valuable comments. We sincerely apologize for incorrectly italicizing M06-2X/6-311++G(2df,2pd). So, the method level has been changed from "M06-2X/6-311++G(*2df,2pd*)" to "M06-2X/6-311++G(2df,2pd)" in the revised manuscript and the revised supporting information. The corresponding main revision has been made as follows.

(a) In Lines 95-96 on Page 4 of the revised manuscript, the method level has been changed from "M06-2X/6-311++G(*2df,2pd*)" to "M06-2X/6-311++G(2df,2pd)".

(b) In Line 110 on Page 4 of the revised manuscript, the method level has been changed from "M06-2X/6-311++G(*2df,2pd*)" to "M06-2X/6-311++G(2df,2pd)".

(c) In Fig. 1 and Table 2 of the revised manuscript, the method level has been changed from "M06-2X/6-311++G(*2df,2pd*)" to "M06-2X/6-311++G(2df,2pd)".

(d) In Fig. S3, Fig. S4, Fig. S12, Table S3 and Table S7 of the revised supporting information, the method level has been changed from "M06-2X/6-311++G(*2df,2pd*)" to "M06-2X/6-311++G(2df,2pd)".

**Comment 9**

Line 142 "Multiwfn 3.8"Actually, there is no Multiwfn 3.8 version. Please check the Multiwfn website and ensure the exact version which the authors downloaded and used, maybe it is Multiwfn 3.8 (dev).

**Response:** We greatly appreciate the reviewer's valuable comments and sincerely apologize for the incorrect citation of the Multiwfn version in our previous work. Following the reviewer's suggestion, we have updated the version of Multiwfn. Now, we confirm that the version utilized in the present study is "Multiwfn 3.8 (dev)". Accordingly, the citation of the Multiwfn version has been corrected to "Multiwfn 3.8 (dev)" in Line 153 on Page 6 of the revised manuscript.

**Comment 10**

Line 151 "Bond lengths were restricted by the LINCS algorithm". Were all the bond lengths in the MD simulations restricted by the LINCS algorithm?

**Response:** Thanks for your valuable comments. As stated by the reviewer, all the bond lengths in the MD simulation need to be constrained by the LINCS algorithm. Consistent with previous studies (*Chemosphere*, 2021, 280, 130709; *Chemosphere*, 2022, 296, 133717; *Sci. Total Environ.*, **2020**, 723, 137987), in the present MD simulation, all the bond lengths were restricted using the LINCS algorithm. So, in Lines 170-171 on Page 6 of the revised manuscript, the sentence of "Bond lengths were restricted by the LINCS algorithm (Hess et al., 1997) to preserve structural integrity during the simulation." has been reorganized as "All the bond lengths were restricted by the LINCS algorithm (Hess et al., 1997) to preserve structural integrity during the simulation."

**Comment 11**

Lines 160-161 "...while $\beta_{i,j}$ stands for the collision rate between $i$ and $j$ clusters. The term $\gamma_{(i+j)\rightarrow i}$ ...". Actually, $\beta_{i,j}$ here is not collision rate but collision coefficient,$\gamma$ is not evaporation rate but evaporation coefficient. The present misleading expression may confuse readership in their understanding of the birth-death equation. Please revise.

**Response:** Thanks for your valuable comments. We sincerely apologize for using wrong misleading expression of the birth-death equation. Indeed, $\beta_{i,j}$ is collision coefficient and $\gamma$ is evaporation coefficient (*Atmos. Chem. Phys.*, 2024, 24, 3593-3612; *Atmos. Environ.,* 2022, 269, 118826; *Environ. Res. Lett.*, 2024, 19, 014076). So, in Lines 179-180 on Page 7 of the revised manuscript, the sentence of "while $\beta_{i,j}$ stands for the collision rate between $i$ and $j$ clusters" has been changed as "while $\beta_{i,j}$ stands for the collision coefficient between $i$ and $j$ clusters". Meanwhile, in Lines 180-181 on Page 7 of the revised manuscript, the sentence of "The term $\gamma_{(i+j)\rightarrow i}\rightarrow i$ refers to the rate at which the larger $i+j$ cluster breaks down (or evaporates) into $i$ and $j$ clusters." has been changed as "The term $\gamma_{(i+j)\rightarrow i}\rightarrow i$ refers to the coefficient at which the larger $i+j$ cluster breaks down (or evaporates) into $i$ and $j$ clusters."

**Comment 12**

Line 167 "...were selected as the boundary clusters in the SA-A-FSA system". Whether a cluster should be selected as a boundary cluster is related to both its evaporation rate and collision rate, as referenced in the article on the ACDC model. Please clarify the selection criteria for boundary clusters in the present study. Furthermore, in lines 313-315, the author pointed out that the evaporation coefficient of $(FSA)_3(A)_3$ $(3.30\times10^{-1}$ s$^{-1})$ was nearly $10^3$ times lower than that of

$(SA)_3(A)_3$ $(2.25\times10^2$ s$^{-1})$. Then why clusters $(FSA)_4(A)_3$, $(FSA)_4(A)_4$, $(FSA)_3(SA)(A)_3$ were not considered as the boundary clusters?

**Response:** Thanks for your valuable comments. Indeed, the boundary conditions require that the outgrowing clusters possess a favorable composition, ensuring their stability and preventing immediate evaporation (***Atom. Chem. Phys.***, **2012**, 12, 9113-9133; ***J. Environ. Sci.***, **2020**, 89, 125-

135). Generally, clusters with low Gibbs free energies and low evaporation rates are considered as suitable boundary clusters. According to the reviewer's suggestion, the Gibbs free energy of formation ($\Delta G$, kcal·mol$^{-1}$) and evaporation rate coefficient ($\gamma$, s$^{-1}$) for the $(FSA)_x(SA)_y(A)_z$ clusters were recalculated firstly based on the energies at the DLPNO-CCSD(T)-F12/cc-pVDZ-F12-

CABS//M06-2X/6-311++G(2df,2pd) level. Then, the clusters of $(SA)_4\cdot(A)_3$, $(SA)_4\cdot(A)_4$,

$(FSA)_4\cdot(A)_3$, $(FSA)_4\cdot(A)_4$, $(FSA)_3\cdot SA\cdot(A)_3$, $(FSA)_2\cdot(SA)_2\cdot(A)_3$ and $FSA\cdot(SA)_3\cdot(A)_3$ were selected as the boundary clusters due to their lower Gibbs free energy and evaporation rates in the SA-A-FSA

system. Notably, the clusters of $(FSA)_4(A)_3$, $(FSA)_4(A)_4$, $(FSA)_3(SA)(A)_3$, as suggested by reviewers, have been incorporated into the newly selected boundary clusters. The corresponding changes are as follows.

(a) According to the reviewer's suggestion, based on the energies at the DLPNO-CCSD(T)-

F12/cc-pVDZ-F12-CABS//M06-2X/6-311++G(2df,2pd) level, the Gibbs free energy of formation ($\Delta G$, kcal·mol$^{-1}$) for the $(FSA)_x(SA)_y(A)_z$ ($z \leq x + y \leq 3$) clusters were recalculated in Fig. 5(a) and

Table S7. Meanwhile, the corresponding evaporation rate coefficient ($\gamma$, s$^{-1}$) for the $(FSA)_x(SA)_y(A)_z$

($z \leq x + y \leq 3$) clusters were recalculated in Fig. 5(b) and Table S8, respectively. The detail revision is provided in the reviewer 2's Comment 4.

(b) Based on the re-calculated Gibbs free energy of formation ($\Delta G$, kcal·mol$^{-1}$) and evaporation rate coefficient ($\gamma$, s$^{-1}$) for the $(FSA)_x(SA)_y(A)_z$ clusters, the clusters of $(SA)_4\cdot(A)_3$, $(SA)_4\cdot(A)_4$,

$(FSA)_4\cdot(A)_3$, $(FSA)_4\cdot(A)_4$, $(FSA)_3\cdot SA\cdot(A)_3$, $(FSA)_2\cdot(SA)_2\cdot(A)_3$ and $FSA\cdot(SA)_3\cdot(A)_3$ were selected as the boundary clusters due to their lower Gibbs free energy and evaporation rates in the SA-A-FSA

system. In Lines 185-187 on Page 7 of the revised manuscript, the sentence of "Therefore, the clusters of $(FSA)_2\cdot(SA)_2\cdot(A)_3$, $(FSA)_1\cdot(SA)_3\cdot(A)_3$, $(SA)_4\cdot(A)_3$ and $(SA)_4\cdot(A)_4$ were selected as the boundary clusters in the SA-A-FSA system." has been changed as "Therefore, the clusters of

$(SA)_4\cdot(A)_3$, $(SA)_4\cdot(A)_4$, $(FSA)_4\cdot(A)_3$, $(FSA)_4\cdot(A)_4$, $(FSA)_3\cdot SA\cdot(A)_3$, $(FSA)_2\cdot(SA)_2\cdot(A)_3$ and

FSA·(SA)$_3$·(A)$_3$ were selected as the boundary clusters in the SA-A-FSA system."

(c) In the section of "3.4 The Impact of Atmospheric Conditions on the Thermodynamic Clusters Stability", the sentence of "The clusters of (SA)$_3$·(A)$_3$, (FSA)$_2$·SA·(A)$_3$ and FSA·(SA)$_2$·(A)$_3$·have the potential to further grow into the boundary clusters [(FSA)$_2$·(SA)$_2$·(A)$_3$, (FSA)$_1$·(SA)$_3$·(A)$_3$ , (SA)$_4$·(A)$_3$ and (SA)$_4$·(A)$_4$]." in Lines 357-360 on Pages 12-13 of the revised manuscript has been changed as "The clusters of (SA)$_3$·(A)$_3$, (FSA)$_3$·(A)$_3$, (FSA)$_2$·SA·(A)$_3$ and FSA·(SA)$_2$·(A)$_3$·have the potential to further grow into the boundary clusters [(SA)$_4$·(A)$_3$, (SA)$_4$·(A)$_4$, (FSA)$_4$·(A)$_3$, (FSA)$_4$·(A)$_4$, (FSA)$_3$·SA·(A)$_3$, (FSA)$_2$·(SA)$_2$·(A)$_3$ and FSA·(SA)$_3$·(A)$_3$], which has relative lower Gibbs free energy and evaporation rates.

**Comment 13**

**Lines 252-255** "In contrast to the SO$_3$ hydrolysis with FSA in the gas phase, which does not proceed within 100 ps, the reaction at the gas-liquid nanodroplet interface rapidly proceeds within just a few picoseconds. This indicates that interfacial water molecules at the gas-liquid nanodroplet interface can accelerate the SO$_3$ hydrolysis.

Q1:Since the FSA catalytic SO$_3$ hydrolysis reaction on the gas-liquid nanodroplet interface is faster, it is more likely to occur on the interface in the case of the low saturated vapor pressure of FSA, making the FSA catalytic reaction in the gas phase relatively less important. And the generated SA molecules at the nanodroplet interface tends to deprotonate rather than evaporate into the gas phase. This is the second contradictory point. Is it possible to calculate the proportion of FSA catalytic reaction occurring in the gas phase versus on the interface? If the proportion of gas phase reaction only accounts for a small fraction, then emphasizing the role of FSA in gas phase reaction and nucleation would appear to have limited significance.

**Response:** Thanks for your valuable comments. Considering the harsh reaction conditions between SO$_3$ and FSA at the interface (i.e., the two molecules must be sufficiently close to formed the SO$_3$-FSA complex), the contribution of the SO$_3$-FSA complex reacting at the aqueous surface is slight due to the low concentration of SO$_3$-FSA complex (ranging from $9.49 \times 10^{-23}$ to $1.80 \times 10^{-22}$ molecules·cm$^{-3}$ within 280.0-320.0 K (Table S2)). Likewise, the hydrolysis of SO$_3$ with FSA in the gas phase, predominantly occurs through collisions between FSA···H$_2$O and SO$_3$ as concentrations of FSA···H$_2$O is $1.36 \times 10^6$-$6.80 \times 10^6$ molecules·cm$^{-3}$ within 280.0-320.0 K, which is at least $10^6$

times larger than those of $SO_3 \cdots H_2O$ and $FSA \cdots SO_3$ (Table S3). Therefore, although the hydrolysis of $SO_3$ with FSA at the gas-liquid nanodroplet interface occurs rapidly within just a few picoseconds and requires less time compared to the gas phase, the hydrolysis of $SO_3$ with FSA predominantly takes place in the gas phase rather than on the aqueous nanodroplet surface. The main explanations are as follows.

(a) For the hydrolysis of $SO_3$ with FSA on the aqueous nanodroplet surface, the interaction between $SO_3$ and FSA at the aqueous interface might take place via three pathways: (*i*) direct interaction of $SO_3$ with adsorbed FSA; (*ii*) interaction of adsorbed $SO_3$ with FSA; or (*iii*) reaction starting from the $SO_3$-FSA complex. Given the high reactivity and the brief residency time of $SO_3$

and FSA at the interface, as evidenced by their short lifetimes (Fig. S8) of only a few picoseconds and rapid formation of $SA^-$ and $FSA^-$ ion, the simulations have primarily considered the pathway of (*iii*). Notably, the contribution of pathway (*iii*) on the aqueous nanodroplet surface is slight due to the low concentration of $SO_3$-FSA complex ($9.49 \times 10^{-23}$-$1.80 \times 10^{-22}$ molecules·cm$^{-3}$ within 280.0-

320.0 K (Table S2)). Notably, the contribution of pathway (*iii*) on the aqueous nanodroplet surface is slight due to the low concentration of $SO_3$-FSA complex ($9.49 \times 10^{-23}$-$1.80 \times 10^{-22}$ molecules·cm$^{-}$

$^3$ within 280.0-320.0 K (Table S2)). However, this focus enabled a deeper understanding of the interfacial dynamics and the mechanisms underpinning these rapid transformations. In Lines 256-

258 on Page 9 of the revised manuscript, the contribution of the $SO_3$-FSA complex reacting on the aqueous surface has been organized as "Notably, the contribution of pathway (*iii*) on the aqueous nanodroplet surface is slight due to the low concentration of $SO_3$-FSA complex ($9.49 \times 10^{-23}$-$1.80$

$\times 10^{-22}$ molecules·cm$^{-3}$ within 280.0-320.0 K (Table S2))".

(b) For the hydrolysis of $SO_3$ with FSA in the gas phase, given the low chance of three molecules of $SO_3$, $H_2O$ and $HCOOSO_3H$ (FSA) colliding simultaneously under atmospheric conditions, the hydrolysis of $SO_3$ with FSA (Channel FSA) was likely a sequential bimolecular process. So, the route for the hydrolysis reaction of $SO_3$ with FSA possibly takes place via FSA +

$SO_3 \cdots H_2O$, $FSA \cdots H_2O + SO_3$ or $FSA \cdots SO_3 + H_2O$ reaction. As the concentration of water molecule ($10^{18}$ molecules·cm$^{-3}$) in the atmosphere is much higher than those of $SO_3$ and FSA ([FSA] = $1.0 \times$

$10^7$, [$SO_3$] = $1.0 \times 10^3$ molecules·cm$^{-3}$) (Liu et al., 2019)), the reaction pathway of $SO_3 \cdots FSA +$

$H_2O$ is hard to occur in actual atmospheric conditions. Under the available concentrations ([FSA] =

$1.0 \times 10^7$, [$SO_3$] = $1.0 \times 10^3$ molecules·cm$^{-3}$) (Liu et al., 2019), the concentration of $FSA \cdots H_2O$ is

$1.36 \times 10^6$-$6.80 \times 10^6$ molecules·cm$^{-3}$ within 280.0-320.0 K, which is $10^6$ times larger than that of

$SO_3$···$H_2O$ (Table S3). So, it is predicted that $SO_3$ hydrolysis with FSA in the gas phase predominantly take places via the collision between FSA···$H_2O$ and $SO_3$. As compared with the

$SO_3$-FSA complex reacting at the aqueous surface, the contribution of FSA···$H_2O$ + $SO_3$ reaction in the gas phase is much larger as the concentration of FSA···$H_2O$ is at least $10^6$ times larger than that of FSA···$SO_3$.

Overall, based on the analysis of reaction mechanisms in the gas phase and on the aqueous nanodroplet surface, we predict that the hydrolysis of $SO_3$ with FSA mainly occurs in the gas phase rather than on the aqueous nanodroplet surface. However, this focus enabled a deeper understanding of the interfacial dynamics and the mechanisms underpinning these rapid transformations.

Q2:Since different molecules in the gas phase can catalyze the $SO_3$ hydrolysis reaction, and water molecules exhibit the fastest catalytic rate due to their higher concentration in the gas phase. So it is necessary to supplement the calculation on the catalytic effect of water molecules on the interface, where their concentration is even higher. A comparison should hence be made between the catalytic effect of water molecules and FSA molecule on the interface.

**Response:** Thanks for your valuable comments. Lv et al. (***Atmos. Environ.***, **2020**, 230, 117514) has investigated the $SO_3$ hydration reaction at the air-water interface, finding that $SO_3$ molecules are efficiently trapped by water droplets and rapidly react to form $HSO_4^-$/$H_3O^+$ or $H_2SO_4$ within a few picoseconds, through a distinct interfacial hydration mechanism compared to the gas-phase reaction.

In contrast, the contribution of FSA-catalyzed $SO_3$ hydrolysis at the gas-liquid nanodroplet interface is relatively slight, as the concentration of water molecules at the aqueous interface is much higher than that of FSA. Based on this analysis, the following sentence has been added in Lines 276-280

on Page 10 of the revised manuscript: "However, considering the harsh reaction conditions between

$SO_3$ and FSA at the interface (i.e., the two molecules must be sufficiently close to formed the $SO_3$-

FSA complex) and the high concentration of water molecules at the aqueous interfaces, the direct hydrolysis of $SO_3$ at the aqueous interfaces is more advantageous than the $SO_3$-FSA complex reacting on the aqueous surface.". In conclusion, the $SO_3$-FSA complex reacting on the aqueous surface is less advantageous than the direct hydrolysis of $SO_3$ at the aqueous interfaces.

**Comment 14**

**Lines 261-264**: "As shown, the interactions of FSA--SA (-21.2 kcal·mol$^{-1}$) and FSA--HNO$_3$ (- 12.1

kcal·mol$^{-1}$) are stronger than that of SA-A (-8.9 kcal·mol$^{-1}$), illustrating that interfacial FSA$^-$ and

H$_3$O$^+$ ions can attract precursor molecules from the gaseous phase to the aqueous nanodroplet surface, and thus facilitating particle growth."

Since the reaction rate for the FSA catalytic SO$_3$ hydrolysis on the aqueous nanodroplet surface is much higher than that in the gas phase. And the generated SA molecules at the nanodroplet interface tends to deprotonate rather than evaporate into the gas phase. Furthermore, interfacial FSA- ions can also attract precursor molecules from the gas phase to the aqueous nanodroplet surface, and thus facilitate particle growth. Combining with the low saturated vapor pressure of FSA, it seems that

FSA is more likely to contribute to the particle growth. In other words, is that possible to calculate the contribution proportion of FSA to the gas phase nucleation and the particle growth? If the contribution proportion of FSA to the gas phase nucleation only accounts for a small fraction, then the FSA-involved nucleation in Section 3.3 and 3.4 would appear to have limited significance.

This is the third contradictory point.

**Response:** Thanks for your valuable comments. Due to the harsh reaction conditions between SO$_3$

and FSA at the interface (i.e., the two molecules must be sufficiently close to formed the SO$_3$-FSA

complex), the contribution of the SO$_3$-FSA complex reacting at the aqueous surface is slight. This is attributed to the low concentration of SO$_3$-FSA complex (ranging from $9.49 \times 10^{-23}$ to $1.80 \times 10^{-22}$

$^{22}$ molecules·cm$^{-3}$ within 280.0-320.0 K). Based on the analysis of the reviewer 2's Comment 13, the hydrolysis of SO$_3$ with FSA predominantly takes place in the gas phase rather than on the aqueous nanodroplet surface. Therefore, we conclude that the contribution of FSA in aerosol nucleation is primarily in the gas phase, rather than at the gas-liquid interface. However, the involvement of FSA$^-$ in aerosol nucleation at the gas-liquid nanodroplet is also important because

FSA$^-$ is expected to demonstrate enhanced nucleation potential at the gas-liquid interface. The main explanations are as follows.

(a) The hydrolysis of SO$_3$ with FSA at the aqueous interface might take place via three pathways: (*i*) direct interaction of SO$_3$ with adsorbed FSA; (*ii*) interaction of adsorbed SO$_3$ with

FSA; or (*iii*) reaction starting from the SO$_3$-FSA complex. Given the high reactivity and the brief residency time of SO$_3$ and FSA at the interface, as evidenced by their short lifetimes (Fig. S6) of only a few picoseconds and rapid formation of SA$^-$ and FSA$^-$ ion, the simulations have primarily considered the pathway of (*iii*). Notably, the contribution of pathway (*iii*) on the aqueous nanodroplet surface is slight due to the low concentration of SO$_3$-FSA complex ($9.49 \times 10^{-23}$-1.80

$\times 10^{-22}$ molecules·cm$^{-3}$ within 280.0-320.0 K (Table S2)). However, this focus enabled a deeper understanding of the interfacial dynamics and the mechanisms underpinning these rapid transformations. Given that the contribution of the hydrolysis of SO$_3$ with FSA on the aqueous nanodroplet surface is slight compared to that in the gas phase, owing to the low concentration of

SO$_3$-FSA complex, we conclude that the contribution of the involvement of FSA$^-$ ions in aerosol nucleation on the aqueous nanodroplet surface is slight, due to the harsh reaction conditions between

SO$_3$ and FSA at the interface.

(b) The hydrolysis of SO$_3$ with FSA in the gas phase possibly takes place via FSA + SO$_3$···H$_2$O,

FSA···H$_2$O + SO$_3$ or FSA···SO$_3$ + H$_2$O reaction. Compared to the SO$_3$-FSA complex reacting at the aqueous surface, the contribution of FSA···H$_2$O + SO$_3$ reaction in the gas phase is much larger as the concentration of FSA···H$_2$O is at least 10$^6$ times larger than that of FSA···SO$_3$. So, we conclude that the hydrolysis of SO$_3$ with FSA predominantly takes place in the gas phase. The involvement of FSA in aerosol nucleation in the gas phase is more significant than its role in aerosol nucleation at the gas-liquid interface.

(c) ACDC kinetic simulations in the gas phase indicated that FSA significantly enhances cluster formation rates in the H$_2$SO$_4$-NH$_3$ system during summer, increasing rates by more than 10$^7$ times under conditions of high FSA concentrations and low H$_2$SO$_4$ and NH$_3$ levels. Meanwhile, the involvement of FSA$^-$ ions in aerosol nucleation at the gas-liquid interface is also importance.

Specially, although the contribution of the hydrolysis of SO$_3$ with FSA at the gas-liquid nanodroplet is slight when it is compared to that in the gas phase, FSA$^-$ is expected to demonstrate enhanced nucleation potential at the gas-liquid interface because the reasons are as follows. The first reason is that the interactions of FSA$^-$-SA (-21.2 kcal·mol$^{-1}$) and FSA$^-$-HNO$_3$ (-12.1 kcal·mol$^{-1}$) are stronger than that of SA-A (-8.9 kcal·mol$^{-1}$), illustrating that interfacial FSA$^-$ and H$_3$O$^+$ ions can attract precursor molecules from the gaseous phase to the aqueous nanodroplet surface, and thus facilitating particle growth. The second reason is that compared to SA-A-*Y* (*Y* = HOOCCH$_2$COOH,

HOCCOOSO$_3$H, CH$_3$OSO$_3$H, HOOCCH$_2$CH(NH$_2$)COOH and HOCH$_2$COOH) , the binding free energy of SA-A-FSA$^-$ (-25.6 kcal·mol$^{-1}$) was larger than 5.2-12.8 kcal·mol$^{-1}$, indicating that the FSA$^-$

at the interface exhibits a greater nucleation capability than gaseous molecule $Y$.

Overall, the hydrolysis of $SO_3$ with FSA mainly occurs in the gas phase rather than at the aqueous nanodroplet surface. The contribution of FSA in aerosol nucleation is significantly greater in the gas phase than at the gas-liquid interface. However, the involvement of $FSA^-$ in aerosol nucleation at the gas-liquid nanodroplet is also important because $FSA^-$ is expected to demonstrate enhanced nucleation potential at the gas-liquid interface.

**Comment 15**

**Line 266:** SA-A-Y ($Y=HOOCCH_2COOH$, $HOCCOOSO_3H$, $CH_3OSO_3H$, $HOOCCH_2CH(NH_2)COOH$ and $HOCH_2COOH$) clusters.

Such comparison does not seem that fair and reasonable. Y and FSA are all acid molecules which can be deprotonated at the nanodroplet interface. So it is more reasonable to compare the binding free energies of $SA-A-FSA^-$ and $SA-A-Y^-$.

**Response:** Thanks for your valuable comments. Y ($Y=HOOCCH_2COOH$, $HOCCOOSO_3H$, $CH_3OSO_3H$, $HOOCCH_2CH(NH_2)COOH$ and $HOCH_2COOH$) has been considered as a sufficient effect on the intermolecular interactions between SA and A molecules and presents a remarkable enhancement effect on specific temperatures and precursor concentration (*Chemosphere*, 2018, 203, 26-33; *J. Phys. Chem. A*, 2020, 124, 3261-3268; *J. Chin. Chem. Soc.*, 2023, 70, 689-698; *Phys. Chem. Chem. Phys.*, 2021, 23, 10184; *J. Chem. Phys.*, 2017, 146, 184308 ). In this work, the apparent nucleation potential of $Y$ molecules in the gas phase for SA-A nucleation is used to qualitatively assess the nucleation potential of $Y^-$ ions at the air-water interface. Strictly speaking, it is methodologically inadequate to use the nucleation potential of gas-phase molecules as a basis for comparing the nucleation potential of ions at the air-water interface. However, in the present study, this assessment is deemed reasonable, as no studies have yet determined the concentration of $Y^-$ ions at the air-water interface. Based on this, we utilized the nucleation potential of gas-phase Y molecules for SA-A aerosols to qualitatively evaluate the nucleation potential for the $Y^-$ ions at the air-water interface. The main explanations are as follows.

(a) Previously reported results suggested that $HOOCCH_2COOH$ (*Chemosphere*, 2018, 203, 26-33) $HOCCOOSO_3H$ (*J. Phys. Chem. A*, 2020, 124, 3261-3268), $CH_3OSO_3H$ (*J. Chin. Chem. Soc.*, 2023, 70, 689-698), $HOOCCH_2CH(NH_2)COOH$ (*Phys. Chem. Chem. Phys.*, 2021, 23, 10184)

and HOCH$_2$COOH (***J. Chem. Phys.***, 2017, 146, 184308) exhibit an excellent nucleation capability on SA-A-driven new particle formation. For example, ACDC simulations indicate that

HOOCCH$_2$CH(NH$_2$)COOH could present an obvious enhancement effect on SA-A-based cluster formation rates, increasing rates by more than $10^4$ times. So, molecules of HOOCCH$_2$COOH,

HOCCOOSO$_3$H, CH$_3$OSO$_3$H, HOOCCH$_2$CH(NH$_2$)COOH and HOCH$_2$COOH are commonly employed to evaluate the contribution of other atmospheric species to the nucleation of SA-A

aerosols.

(b) Binding free energies have been widely used to qualitatively assess the potential of ion in aerosol nucleation (***Angew. Chem. Int. Ed.***, **2019**, 58, 8351-8355; ***Atmos. Chem. Phys.***, 2024, 24,

4029-4046). This evaluation method offers a qualitative assessment of the nucleation potential of ions at the interface, rather than a precise quantitative analysis. A more detailed quantitative evaluation would necessitate data on the Y ion concentration at the interface; however, no such measurements have been reported. Therefore, to assess the aerosol nucleation potential of FSA$^-$ ions at the gas-liquid interface, we compared their binding energies with SA-A to those of Y molecules with SA-A.

(c) Based on this analysis, the sentence has been added in Lines 297-299 on Pages 10-11 of the revised manuscript: "A further quantitative assessment of the aerosol nucleation potential of Y ions at the droplet interface could not be conducted, as data on the concentration of Y ions at the interface are not yet available."

**Comment 16**

Line 283 "In these simulation systems, 5 FSA, 5 SA, 10 A, 20 H$_2$O, 41 O$_2$ and 154 N$_2$ molecules were included."

The ratio of the number of molecules does not seem to be consistent with the real atmospheric concentration condition. Once the number of molecules in the Gromacs MD simulation is large enough, molecular aggregation will take place.

**Response:** Thanks for your valuable comments. MD simulations has effectively assessed the nucleation potential of the product from the reactions between Criegee intermediates and atmospheric trace species (***Chemosphere***, 2022, 296, 133717; ***Atmos. Environ.***, 2024, 330, 120558;

***Sci. Total Environ.***, 2020, 723, 137987). Consistent with the previously studies (***Chemosphere***,

2021, 280, 130709; ***Int. J. Mol. Sci.***, 2023, 24, 5400; ***Atmos. Environ.***, 2024, 320, 120338), each simulation systems comprised 5 FSA, 5 SA, 10 A, 20 $H_2O$, 41 $O_2$ and 154 $N_2$ molecules. Besides, similar with the previously studies (***Chemosphere***, 2022, 296, 133717; ***Atmos. Environ.***, 2024, 330,

120558; ***Sci. Total Environ.***, 2020, 723, 137987; ***Chemosphere***, 2021, 280, 130709; ***Int. J. Mol.***

***Sci.***, 2023, 24, 5400; ***Atmos. Environ.***, 2024, 320, 120338 ), the concentration of precursors has not been considered, and only a qualitative assessment of FSA's involvement in SA-A nucleation was conducted. So, the sentence of "Similar with the previously studies (Ding et al., 2024; Wei et al.,

2022; Li et al., 2023), the concentration of precursors has not been considered, and only a qualitative assessment of FSA's involvement in SA-A nucleation was conducted." has been added and organized in the Lines 310-313 on Page 11 of the revised manuscript.

**Comment 17**

**Fig.4.**

The classical MD simulation performed by Gromacs relies on the force field of the molecules, which can only describe physical aggregation. However, during the nucleation process, the proton transfer between acid and base molecules plays an important role in acid-base nucleation which cannot be reflected in the classical MD simulation. In this case, the physical aggregation pathways in the MD simulations are not completely consistent with those in ACDC simulations.

For example, in the MD simulation shown in Fig. 4, all the clusters containing SA, A and FSA.

In the ACDC simulations, there exist pure SA-A paths. The clusters growth pathways are not completely consistent with each other. This is the fourth contradiction in this manuscript which may make the readership confused.

**Response:** Thanks for your valuable comments. Indeed, in this work, the atmospheric implications and effect mechanism of FSA in the SA-A-dominated NPF process were evaluated both qualitatively and quantitatively as follows. Firstly, to investigate the aggregation trends of FSA with

SA and A qualitatively, the classical MD simulations were employed to observe how FSA aggregates with SA and A at three different temperatures of 258.15 K, 278.15 K and 298.15 K (Fig. 4 and Fig.

S10-S11). During the clustering process at three different temperatures of 258.15 K, 278.15 K and

298.15 K, FSA attracted SA and A molecules, thereby facilitating the formation of larger clusters, with hydrogen bonding playing a critical role in these interactions. It is noteworthy that the concentration of precursors has not been considered and only a qualitative assessment of FSA's involvement in SA-A nucleation was conducted. It is also noteworthy that during the nucleation process, the proton transfer between acid and base molecules plays an important role in acid-base nucleation which cannot be reflected in the classical MD simulation. However, it is initially predicted by classical MD simulation that FSA could act as a "participator" in NPF and could be directly involved in SA-A nucleation. Further predictions regarding the enhancement effect of FSA on SA-A molecular clustering should be conducted below by considering the cluster stability, the formation rate and the growth pathways. So, the limitation of classical molecular dynamic (MD) simulations has been added in Lines 325-331 on Pages 11-12 of the revised manuscript.

Based on the qualitative MD simulations outlined above, the enhancement effect and NPF mechanism of FSA under different temperatures and precursor concentrations were quantitatively studied using the quantum chemical calculation combined with Atmospheric Cluster Dynamics Code (ACDC). The quantitative analysis shows that, throughout the clustering process, FSA was observed to attract SA and A molecules, thereby facilitating the formation of larger clusters, with hydrogen bonding playing a critical role in their interactions. These findings by the quantitative results further support the qualitative MD predictions, reinforcing the notion that FSA acts as a "participant" in the NPF process. So, the sentence about ACDC simulation results validating simple predictions of classical molecular dynamics was reorganized in Lines 382-385 on Page 13 of the revised manuscript.

In addition to the qualitative predictions obtained from MD simulations, the quantitative result from quantum chemical calculation combined with the ACDC further indicate the three additional results as follows.

(*i*) Protonation, analyzed via the localized orbital locator (LOL) distribution (Schmider et al., 2000) and the Laplacian bond order (LBO), are commonly used to qualitatively assess the impact of atmospheric trace species on the stability of acid-base binary systems. As shown in Fig. S13, in contrast to the pure SA-A cluster, where proton transfer from SA to A has not occurred, the proton in the SA·A·FSA cluster is fully transferred from FSA to A. This is evidenced by the green region between H1 and O1 and the red region between H1 and N1 in Fig. S13(b), and the LBO of the newly formed H1···N1 bond in the SA·A·FSA cluster is 0.606 a.u. Similar to the SA·A·FSA cluster, the MA in the SA·A·$(FSA)_2$ cluster is protonated (Fig. S13(b)), but to a greater extent. These analyses demonstrate that the FSA molecule not only strengthens the hydrogen bonding between SA and A

but also facilitates proton transfer from SA to A, thereby enhancing the stability of the SA-A cluster.

So, Fig. S13 and Table S6 have been added in the revised supporting information.

(*ii*) The values of $J$ for the SA-A-FSA system at varying temperatures (Fig. 6) showed that $J$

increased as the temperature decreased, due to the smaller values of both $\Delta G$ and $\gamma$ at lower temperatures. Specifically, when [FSA] ranges from $10^3$ to $10^7$ molecules·cm$^{-3}$, $J$ can increase by up to four orders of magnitude at 258.15 K. At 298.15 K, $J$ shows a significant increase, rising by five orders of magnitude. These findings suggest that the formation rate exhibits a substantial variation at high temperatures.

(*iii*) Precursor concentration is a significant factor influencing the nucleation process of the

SA-A-based system. Specifically, $J$ increased with increasing [FSA], attributable to the formation of more SA-A-FSA clusters. For example, when [FSA] exceeds $10^3$ molecules·cm$^{-3}$ at the high temperature of 298.15 K, $J$ exhibits a significant increase, rising by five orders of magnitude. This suggests that the involvement of FSA can strongly enhance the nucleation rate in SA-A-based NPF.

In addition to temperature and [FSA], the varying concentrations of SA and A might have a significant impact on the nucleation rate. Fig. 7 reveals a clear positive correlation between $J$ and both [SA] and [A]. This can also be attributed to the fact that a higher concentration of nucleation precursors promotes an increase in $J$.

So, the quantitative analysis by ACDC simulations not only confirmed the qualitative predictions of classical MD simulations but also revealed the effect of proton transfer, temperatures and concentrations on SA-A nucleation. The main revisions are as follows.

(a) In the revised manuscript of Fig. 4, we have added the pure SA-A clusters. The newly revised Fig. 4 is shown below.

[Figure]

**Fig. 4.** Snapshots of nucleation simulation at 258.15 K from FSA, SA and A using the VDW representation, with $N_2$ and $O_2$ shown using the line drawing method.

(b) In the section of "3.3 FSA's Role in Nucleation and Cluster Formation", we reanalyzed the aggregation process of MD simulations, the sentence of "Subsequently, FSA can bind with SA and A to form FSA·A, FSA·SA·A and FSA·SA·$(A)_3$ clusters at 1.5 ns, and then the FSA·SA·A, $(FSA)_2$·SA·$(A)_3$ and $(FSA)_2$·$(SA)_2$·$(A)_3$ clusters are formed at 3.0 ns. Next, with further aggregation of FSA molecules, $(FSA)_2$·SA·$(A)_4$ and $(FSA)_3$·$(SA)_3$·$(A)_4$ clusters are observed within 4.0 ns. Finally, the FSA molecules fully aggregate to form $(FSA)_5$·$(SA)_5$·$(A)_{10}$ clusters at 7.5 ns, and this complete cluster stays stable throughout the entire simulation period." in Lines 316-323 on Page 11 of the revised manuscript has been changed as "Subsequently, at 0.4 ns, various clusters such as SA·A and FSA·A clusters were formed. As molecular aggregation continued, the collision between FSA, SA, and A molecules results in the formation of SA·$(A)_2$, FSA·A, FSA·SA·A and FSA·SA·$(A)_3$ clusters at 1.5 ns, and then the SA·$(A)_2$, FSA·SA·A, $(FSA)_2$·SA·$(A)_3$ and $(FSA)_2$·$(SA)_2$·$(A)_3$ clusters are formed at 3.0 ns. Next, with further aggregation of the molecules, SA·$(A)_2$, $(FSA)_2$·SA·$(A)_4$ and $(FSA)_3$·$(SA)_3$·$(A)_4$ clusters are observed within 4.0 ns. Finally, the molecules fully aggregate to form $(FSA)_5 \cdot (SA)_5 \cdot (A)_{10}$ clusters at 7.5 ns, and this complete cluster stays stable throughout the entire simulation period."

(c) The limitation of classical molecular dynamic (MD) simulations has been added in Lines

325-331 on Pages 11-12 of the revised manuscript, which has been organized as "It is also noteworthy that during the nucleation process, the proton transfer between acid and base molecules plays an important role in acid-base nucleation which cannot be reflected in the classical MD

simulation. However, it is initially predicted by classical MD simulation that FSA could act as a

"participator" in NPF and could be directly involved in SA-A nucleation. Further predictions regarding the enhancement effect of FSA on SA-A molecular clustering should be conducted below by considering the cluster stability, the formation rate and the growth pathways."

(d) Fig. S13 and Table S6 have been added in the revised supporting information.

(e) The sentence about ACDC simulation results validating simple predictions of classical molecular dynamics was reorganized in Lines 382-385 on Page 13 of the revised manuscript, which has been added as "The SA-A-FSA nucleation pathway can be categorized into two routes, with

FSA acting as a "participator" in the SA-A-FSA-based nucleation process. This is in agreement with the results predicted by the molecular dynamics (MD) simulations."

**Comment 18**

**Line 352:** "First, as [A] increases, the interaction between FSA and SA in the ternary cluster may be disrupted, leading to a decrease in the saturation of FSA interaction sites and a reduction in R."

According to the data shown in Fig.5, the $\Delta G$ and evaporation coefficient of FSA-based clusters is much lower than that of SA-based clusters. As [A] increases, if the interaction between FSA

and SA in the ternary cluster was disrupted, the FSA will interact with the increasing A. The generated FSA-A-based clusters are more thermodynamically stable, and hence the R will increase. This is somewhat inconsistent with the current conclusion.

**Response:** Thank you for your valuable comments. Based on the energies at the DLPNO-CCSD(T)-

F12/cc-pVDZ-F12-CABS//M06-2X/6-311++G(2df,2pd) level, the Gibbs free energy of formation ($\Delta G$, kcal·mol$^{-1}$) for the $(FSA)_x(SA)_y(A)_z$ ($z \leq x + y \leq 3$) clusters were recalculated. A range of

ACDC simulations were performed using the updated thermodynamic data for the SA-A-FSA

clusters at various temperatures and monomer concentrations. Since the value of enhancement factor ($R$) is defined as $R=J_{SA\text{-}A\text{-}FSA}/J_{SA\text{-}A}$, it represents a relative ratio rather than an absolute value.

So, the $R$ is ineffective in evaluating FSA impact on cluster nucleation under different atmospheric conditions. In some special cases, the conclusions obtained based on the $R$ are wrong. For example, as the temperature rises, the increase of $J_{SA\text{-}A}$ is larger than that of $J_{SA\text{-}A\text{-}FSA}$, implying a decrease in the value of $R$. On the contrary, as the temperature decreases, the decrease of $J_{SA\text{-}A}$ is smaller than the corresponding of $J_{SA\text{-}A\text{-}FSA}$, suggesting an increase in the value of $R$. So, $J$ of SA-A-FSA-based system was mainly discussed, rather than the $R$. This situation has been found in the previous references (**Atmos. Chem. Phys.**, 2024, 24, 5823-5835; **Environ. Sci. Technol**., 2024, 58, 16962-

16973; **Atmos. Environ.**, 2024, 318, 120266). Based on these analysis, the influence of temperature and the precursor concentration on the formation rate ($J$, cm$^{-3}$·s$^{-1}$) has been further analyzed and reorganized in the revised manuscript. The corresponding revision has been mainly made as follows.

(a) Based on the energies at the DLPNO-CCSD(T)-F12/cc-pVDZ-F12-CABS//M06-2X/6-

311++G(2df,2pd) level, Gibbs free energy of formation ($\Delta G$, kcal·mol$^{-1}$), evaporation rate coefficient ($\gamma$, s$^{-1}$), collision coefficients ($\beta$, cm$^3$ s$^{-1}$), total evaporation coefficients ($\sum\gamma$, s$^{-1}$) and ratios ($\beta C/\sum\gamma$) of monomer collisions of the (FSA)$_x$(SA)$_y$(A)$_z$ clusters were recalculated, which were reorganized in revised Fig. 5 and Table S7-S11. Meanwhile, in the whole revised manuscript, the values of Gibbs free energy of formation ($\Delta G$, kcal·mol$^{-1}$) and evaporation rate coefficient ($\gamma$, s$^{-1}$)

for the (FSA)$_x$(SA)$_y$(A)$_z$ ($z \leq x + y \leq 3$) clusters have been updated.

(b) A range of ACDC simulations were performed using the updated thermodynamic data for the SA-A-FSA clusters at various temperatures and monomer concentrations. The cluster formation rate ($J$, cm$^{-3}$·s$^{-1}$) of the SA-A-FSA clusters have been recalculated at varying temperatures and monomer concentrations, which has been reorganized in revised Fig. 6, Fig. 7 and Table S12-S14.

(c) The value of $R$ is defined as $R=J_{SA\text{-}A\text{-}FSA}/J_{SA\text{-}A}$, it represents a relative ratio rather than an absolute value. So, the formation rate ($J$, cm$^{-3}$·s$^{-1}$), rather than the enhancement factor, has been used to effectively assess the influence on the new particle formation. Consistent with the previously studies (**Atmos. Chem. Phys.**, 2024, 24, 5823-5835; **Environ. Sci. Technol**., 2024, 58, 16962-16973;

**Atmos. Environ.**, 2024, 318, 120266), the formation rate ($J$, cm$^{-3}$·s$^{-1}$) of the system was mainly discussed, rather than the enhancement factor. Based on this, in the section of "3.5 Influence of

Particle Formation Rates Under Varying Temperatures and Nucleation Precursor Concentrations"

of the revised manuscript, the analysis of the influence of temperature and nucleation precursor concentrations on formation rate ($J$, cm$^{-3}$·s$^{-1}$) has been mainly analyzed. In the Lines 368-378 on Page 14 of the revised manuscript, the sentence of "Specifically, when [FSA] ranges from $10^3$ to $10^7$ molecules·cm$^{-3}$, $J$ can increase by up to four orders of magnitude at 258.15 K. At 298.15 K, $J$ shows a significant increase, rising by five orders of magnitude. These findings suggest that the formation rate exhibits a substantial variation at high temperatures. Meanwhile, $J$ increased with increasing [FSA], attributable to the formation of more SA-A-FSA clusters. For example, when [FSA] exceeds $10^3$ molecules·cm$^{-3}$ at the high temperature of 298.15 K, $J$ exhibits a significant increase, rising by five orders of magnitude. This suggests that the involvement of FSA can strongly enhance the nucleation rate in SA-A-based NPF. In addition to temperature and [FSA], the varying concentrations of SA and A might have a significant impact on the nucleation rate. Fig. 7 reveals a clear positive correlation between $J$ and both [SA] and [A]. This can also be attributed to the fact that a higher concentration of nucleation precursors promotes an increase in $J$." has been added and organized.

(d) Based on the Gibbs free energy of formation at the DLPNO-CCSD(T)-F12/cc-pVDZ-F12-CABS//M06-2X/6-311++G(2df,2pd) level, the growth pathways and contributions of the SA-A-FSA clusters have been recalculated at varying temperatures and monomer concentrations, and these results have been reorganized in Fig. 8-9 of the revised manuscript. In the section of "3.6 FSA-Driven Nucleation Enhancement Mechanism" of the revised manuscript, the analysis of the influence of the growth pathways and contribution of the SA-A-FSA clusters has been reorganized. In the Lines 385-407 on Pages 13-14 of the revised manuscript, the sentence of "One route involved the initial formation of the stable cluster FSA·A, which then collided with one FSA molecule or another FSA·A cluster to form subsequent stable clusters and continue growing. The other route involved the initial formation of the stable $(SA)_2$·A cluster, which then collided with one FSA·A cluster to form the stable $(SA)_2$·$(A)_2$·FSA, continuing to grow through the addition of an A molecule. Interestingly, at varying temperatures and concentrations of nucleating precursors, the FSA molecule exhibited distinct effects and contributions in the SA-A system. As the temperature increased, the contribution of the SA-A-FSA pathway rose from 6% to 92% (Fig. 9(a)). Therefore, the cluster growth pathway involving FSA appears to prevail at relatively higher temperatures, such as during summer or at lower altitudes. The involvement of FSA in the primary cluster formation pathway may also be influenced by the concentration of the precursors. Specifically, the contribution of the FSA participation pathway exhibited a negative correlation with [SA] or [A] at 278.15 K (Fig. 9(b-c)). Consequently, the contributions of the SA-A-FSA pathway may be more substantial in the clean atmospheric boundary layer with low [A] and [SA], such as in area distant from heavy traffic and emission sources of SA. Additionally, the contribution of the SA-A-FSA pathway increases as [FSA] rises (Fig. 9(d)). At lower [FSA] ($10^4$ molecules·cm$^{-3}$), the contribution of SA-A-FSA pathway was only 15%, with cluster growth pathways predominantly governed by the formation of pure SA-A clusters. However, as [FSA] increased to $10^5$ molecules·cm$^{-3}$, the contribution of FSA-involved clusters rose to 64%, making the pathway involving FSA dominant for cluster formation in the SA-A-FSA system. Moreover, the SA-A-FSA mechanism contributed more significantly (94%) at higher [FSA] concentrations ($10^6$-$10^7$ molecules·cm$^{-3}$). In summary, the contribution of the pathway involving FSA is significantly prevalent in the NPF process with decreasing [SA] and [A] and increasing temperature and [FSA]. These results suggest that FSA could be a significant contributor to SA-A atmospheric NPF, and the SA-A-FSA pathway may prevail in regions with relatively higher temperatures and high FSA emissions, such as in Beijing, Shanghai, and Tangshan, where high concentrations of SO$_3$ and HCOOH are observed." has been added and organized.

Overall, since the value of $R$ is defined as $R=J_{SA-A-FSA}/J_{SA-A}$, the formation rate ($J$, cm$^{-3}$·s$^{-1}$) of SA-A-FSA-based system was mainly discussed, rather than the enhancement factor.

**Comment 19**

**Line 386**: "...with high FSA emissions..."

Is there any emissions sources of FSA?

**Response:** Thanks for your valuable comments. The emission sources of FSA in the atmosphere have not yet been reported directly. However, carboxylic sulfuric anhydrides (CSAs) are a recently identified class of atmospheric organosulfides, formed by the cycloaddition of SO$_3$ with organic carboxylic acids present (Fleig et al., 2012). Specifically, FSA is produced via the addition reaction between SO$_3$ and HCOOH. Based on this reaction, we predict that the primary sources of FSA emissions will be regions with high concentrations of SO$_3$ and HCOOH, such as Beijing, Shanghai, and Tangshan. It is important to note that the emission sources of FSA were predicted based on the reaction between SO$_3$ and FA in the atmosphere. To accurately determine the emission sources of FSA, further extensive field observations are necessary for a more comprehensive investigation. So, in the Lines 407-410 on Page 14 of the revised manuscript, the sentence of "These results suggest that FSA could be a significant contributor to SA-A atmospheric NPF, and the SA-A-FSA pathway may prevail in regions with relatively higher temperatures and high FSA emissions, such as in Beijing, Shanghai, and Tangshan, where high concentrations of $SO_3$ and HCOOH are observed." has been organized.

**Comment 20**

**Line 592**: Smith, C. J., Huff, A. K., Ward, R. M., and Leopold, K. R.: Carboxylic sulfuric anhydrides, J. Phys. Chem. A, 124, 601-612, https://doi.org/10.1126/science.1180315, 2020. The URL (https://doi.org/10.1126/science.1180315) provided does not match this reference, the author wrote a wrong URL. Please revise and also check the whole references section.

**Response:** Thanks for your valuable comments. We are very sorry for using the wrong URL for the reference to Carboxylic sulfuric anhydrides. The correct reference have been recited and organized as, "Smith, C. J., Huff, A. K., Ward, R. M., and Leopold, K. R.: Carboxylic sulfuric anhydrides, J. Phys. Chem. A, 124, 601-612, https://doi.org/ 10.1021/acs.jpca.9b09310, 2020." Simultaneously, we rechecked all the references in the manuscript to ensure that they were cited correctly.

**Technical corrections:**

(1) Line 72: A full stop "." should be added at the end of the paragraph.

**Response:** Thanks for your valuable comments. According to the reviewer's suggestion, the sentence of "Thus, it is essential to investigate whether FSA accelerates $SO_3$ hydrolysis at the gas-liquid nanodroplet interface, as this could offer valuable insights into atmospheric chemistry and the mechanisms driving particle formation" in Lines 68-70 on Page 3 of the revised manuscript has been changed as "Thus, it is essential to investigate whether FSA accelerates $SO_3$ hydrolysis at the gas-liquid nanodroplet interface, as this could offer valuable insights into atmospheric chemistry and the mechanisms driving particle formation."

(2) Line 102: "The most stable structure of the $(FSA)_x(SA)_y(A)_z$ ($z \leq x + y \leq 3$)...".This is the first time that abbreviation "A" appeared. It will be clearer to clarify that "A" is the abbreviation of NH$_3$.

**Response:** Thanks for your valuable comments. According to the reviewer's suggestion, the first time sulfuric acid is mentioned in Line 34 on Page 2 of the revised manuscript, the abbreviation "SA" will be used; similarly, the first time ammonia is mentioned in Line 73 on Page 3 of the revised manuscript, the abbreviation "A" will be used.

(3) Results and discussion: The tense of the sentences throughout the "Results and discussion" should be consistent. If the past tense is used, please use the past tense uniformly. For example, in lines 173 and 176, the authors used "was" and "is", respectively.

**Response:** Thanks for your valuable comments. According to the reviewer's suggestion, the entire "Results and Discussion" section has been reviewed to ensure consistency in the tense throughout the revised manuscript.

(4) Line 376: "low"→"lower". line 386: "high"→ "higher".

**Response:** Thanks for your valuable comments. According to the reviewer's suggestion, in Lines 400 on Page 14 of the revised manuscript, we have revised "low" to "lower". Meanwhile, in Lines 408 on Page 14 of the revised manuscript, we have revised "high" to "higher".

(5) Fig.4: "VDW"→"vdW".

**Response:** Thanks for your valuable comments. According to the reviewer's suggestion, we have revised "VDW" to "vdW".

(6) Fig.7: "$T$=278 K"→"$T$=278.15 K"

**Response:** Thanks for your valuable comments. According to the reviewer's suggestion, we have revised the temperature in Fig. 7 from 278 K to 278.15 K.